# Sample Complexity of Branch-length Estimation by Maximum Likelihood

**David Clancy, Jr.** [* 1]  **Hanbaek Lyu** [* 1]  **Sebastien Roch** [* 1]

## Abstract

We consider the branch-length estimation problem on a bifurcating tree: a character evolves along the edges of a binary tree according to a two-state symmetric Markov process, and we seek to recover the edge transition probabilities from repeated observations at the leaves. This problem arises in phylogenetics, and is related to latent tree graphical model inference. In general, the log-likelihood function is non-concave and may admit many critical points. Nevertheless, simple coordinate maximization has been known to perform well in practice, defying the complexity of the likelihood landscape. In this work, we provide the first theoretical guarantee as to why this might be the case. We show that deep inside the Kesten-Stigum reconstruction regime, provided with polynomially many $m$ samples (assuming the tree is balanced), there exists a universal parameter regime (independent of the size of the tree) where the log-likelihood function is strongly concave and smooth with high probability. On this high-probability likelihood landscape event, we show that the standard coordinate maximization algorithm converges exponentially fast to the maximum likelihood estimator, which is within $O(1/\sqrt{m})$ from the true parameter, provided a sufficiently close initial point.

## 1. Introduction

Maximum likelihood estimation (MLE) for parameter estimation is a fundamental technique in statistics and machine learning. For this, one has a parametric probabilistic model $(\mathbb{P}_{\boldsymbol{\theta}}; \boldsymbol{\theta} \in \Theta \subset \mathbb{R}^d)$ and some data $x_1, \cdots, x_m$ assumed to be i.i.d. observations from $\mathbb{P}_{\boldsymbol{\theta}^*}$ for some unknown $\boldsymbol{\theta}^* \in \Theta$.

*Equal contribution [1]Department of Mathematics, University of Wisconsin-Madison, WI, USA. Correspondence to: David Clancy, Jr. <dclancy@math.wisc.edu>, Hanbaek Lyu <hlyu@math.wisc.edu>, Sebastien Roch <roch@math.wisc.edu>.

*Proceedings of the 42nd International Conference on Machine Learning*, Vancouver, Canada. PMLR 267, 2025. Copyright 2025 by the author(s).

One seeks an estimate $\hat{\boldsymbol{\theta}} \in \Theta$ such that

$$\hat{\boldsymbol{\theta}} \in \arg\max_{\boldsymbol{\theta} \in \Theta} \ell(\boldsymbol{\theta}; x_1, \ldots, x_m), \text{ where}$$

$$\ell(\boldsymbol{\theta}; x_1, \ldots, x_m) = \frac{1}{m} \sum_{j=1}^{m} \log \mathbb{P}_{\boldsymbol{\theta}}(x_j), \qquad (1)$$

which maximizes the likelihood of seeing the observed data.

Classical theory (Cramér, 1946; Wald, 1949) tells us that whenever the population landscape (i.e. the $m \to \infty$ limit)

$$\mathbb{E}[\ell(\boldsymbol{\theta})] = \mathbb{E}_{X \sim \mathbb{P}_{\boldsymbol{\theta}^*}} \log \mathbb{P}_{\boldsymbol{\theta}}(X)$$

is concave and the likelihood is maximized at a unique point that agrees with the true parameter $\boldsymbol{\theta}^*$, then $\hat{\boldsymbol{\theta}}$ is a good estimator of $\boldsymbol{\theta}^*$. Much of the rigorous theoretical foundation of MLE assumes that one can extend this population landscape to the empirical landscape $\ell(\boldsymbol{\theta}; x_1, \ldots, x_m)$. This relies on being able to approximate $\mathbb{E}[\ell(\boldsymbol{\theta})]$ by $\ell(\boldsymbol{\theta}; x_1, \cdots, x_m)$ so that optimizing $\ell(\boldsymbol{\theta}; x_1, \ldots, x_m)$ approximately finds $\boldsymbol{\theta}^*$. See (Li & Babu, 2019). However, many natural MLE problems such as those arising from mixture models (Murphy, 2012) involve maximizing non-concave log-likelihoods.

A similar story arises in the optimization literature, where one typically minimizes a (hopefully convex) loss function. Many natural questions on that side are non-convex due to either the loss function being non-convex or having to optimize over a non-convex parameter space, such as principal component analysis and best subset selection problems (Beale et al., 1967; Hocking & Leslie, 1967). Within the class of non-convex problems, there are many examples of *benign non-convex* problems which, while not actually convex, possess sufficiently nice structures that make them manageable. These include some additional symmetries, such as those in phase retrieval (Sun et al., 2018) and dictionary learning (Arora et al., 2014; Bai et al., 2019), or the loss function possesses sufficient regularity (e.g. the Polyak-Łojasiewicz (Polyak, 1963; Lojasiewicz, 1963) condition) so that gradient-based methods converge linearly despite non-convexity (Karimi et al., 2016). See (Jain et al., 2017; Zhang et al., 2020) for a general overview.

Here we investigate a specific maximum likelihood estimation problem along the following lines. Even though the population and empirical landscapes may be non-concave

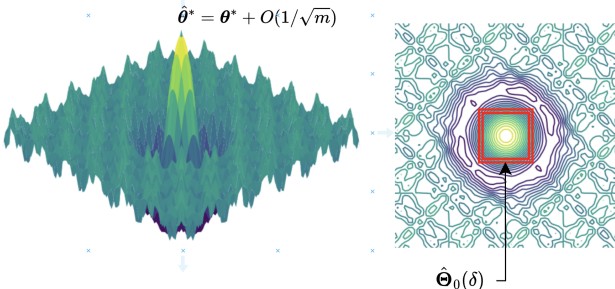

$$\hat{\boldsymbol{\theta}}^* = \boldsymbol{\theta}^* + O(1/\sqrt{m})$$

$$\hat{\boldsymbol{\Theta}}_0(\delta)$$

Figure 1. A cartoon depiction of non-concave 2D likelihood landscape (left) and its contour plot (right). Thm. 3.2 and 3.3 asserts that the empirical likelihood landscape has a well-conditioned strongly concave landscape over a box $\widehat{\boldsymbol{\Theta}}_0(\delta)$ of size order 1 around the true parameter $\boldsymbol{\theta}^*$. Thm. 3.4 asserts that coordinate maximization initialized in the box converges to the MLE $\hat{\boldsymbol{\theta}}_0$ in $O(1)$ iterations.

and contain numerous local (or even global) maximizers, under particular assumptions one can still extract a semi-global region inside which there is a unique maximizer and all other local maximizers lie strictly outside this region. This semi-global region is one that does not shrink as the sample size or the problem dimension grows large, but is not the entire parameter space $\boldsymbol{\Theta}$ either. If this is the case, then one should be able to recover the true parameter $\boldsymbol{\theta}^*$ using standard likelihood maximization techniques and assess the incurred statistical and computational errors rigorously.

More specifically, along this line of approach, we analyze the MLE problem for *branch-length estimation* (Guindon & Gascuel, 2003), arising in phylogenetics, under the Cavender-Farris-Neyman model (Neyman, 1971; Farris, 1973; Cavender, 1978). Roughly speaking, it is the problem of estimating the probabilities of corruption in a noisy channel along the edges of a communication tree, where we are only allowed to observe signals at the tips of the tree (see Sec. 2). While arising in phylogenetics, it has applications to theoretical computer science, signal processing, and statistical physics (Mossel, 2022). This problem is hard as the population log-likelihood function admits exponentially many critical points (in the problem dimension) and there are instances where the empirical log-likelihood has multiple global maximizers. We work deep inside the Kesten-Stigum reconstruction regime (Kesten & Stigum, 1966; Bleher et al., 1995; Ioffe, 1996) where the corruption probabilities are sufficiently small so that signal at internal nodes can be approximated with good accuracy from just observing the signals at the leafs.

### 1.1. Contribution

For this branch length estimation problem, we establish the following results under some assumptions stated later on (see Fig. 1 for illustrations):

- (*Empirical likelihood landscape*, Thm. 3.2) With enough samples $m$ (polynomial in the size of the tree in the balanced case), the empirical log-likelihood is strongly concave and smooth on an $L_\infty$ box around the true parameter $\boldsymbol{\theta}^*$ with large probability.

- (*Statistical estimation guarantee*, Thm. 3.3) For any fixed problem size, the MLE is $O(1/\sqrt{m})$-consistent with the true parameter with arbitrarily large probability.

- (*Computational guarantee of coordinate maximization*, Thm. 3.4) The iterates of the coordinate maximization algorithm (Alg. 1) converge exponentially fast to the confined MLE $\hat{\boldsymbol{\theta}}^*$ with a rate independent of the tree, provided a sufficiently close initial point.

While we focus on a particular likelihood function and a particular optimization algorithm, our general approach relies on three essential ingredients for analyzing such non-concave MLE problems. Namely:

Step 1. Show that the population likelihood $\mathbb{E}[\ell(\boldsymbol{\theta})]$ is strongly concave and smooth on some parameter space $B \subseteq \boldsymbol{\Theta}$ containing the true parameter $\boldsymbol{\theta}^*$;

Step 2. Show that entries of the population Hessian vary in a Lipschitz manner with respect to the parameter; and

Step 3. Show that the per-sample empirical Hessian has uniformly bounded spectral norm almost surely.

These conditions are enough to show that, with arbitrarily large probability, with enough samples $m$, the empirical likelihood landscape concentrates around the population likelihood landscape on $B$ with high probability. Hence, on this high-probability event, the empirical likelihood landscape restricted on $B$ looks 'benign', and standard algorithms for computing MLE, initialized in $B$, converges rapidly to a parameter $\hat{\boldsymbol{\theta}}$ that is within $O(1/\sqrt{m})$ from the true parameter. In short, the Lipschitz condition on the entries of the Hessian allows us to control the fluctuations of the eigenvalues of the Hessian in a non-trivial region around the true parameter. One can then use a uniform version of matrix concentration (Lemma 4.5) to turn this local behavior at points to the desired semi-global statement around $\boldsymbol{\theta}^*$.

### 1.2. Related Works

The model we analyze is the Cavender-Farris-Neyman (CFN) model (Neyman, 1971; Farris, 1973; Cavender, 1978) used to study molecular evolution along an evolutionary tree.

It has been a long-standing open problem to establish rigorously conditions under which standard gradient/coordinate descent algorithms for the maximum-likelihood principle can solve the branch-length estimation problem (Felsenstein,

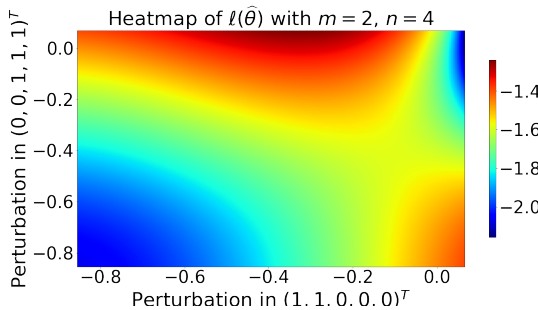

*Figure 2.* A 2D slice of a 5D 2-sample empirical log-likelihood $\ell(\hat{\boldsymbol{\theta}}; \sigma^{(1)}, \sigma^{(2)})$ containing $\boldsymbol{\theta}^*$ on a tree with $n = 4$ leaves.

1981). The corresponding joint likelihood function in general is non-concave, admitting potentially many (complex) critical points (García Puente et al., 2024) for generic data $\sigma^{(j)}$. Notably, Steel (1994) provided an explicit example where there are multiple global maximizers to (2) with two samples on the tree with four leaves, two internal nodes, and five edges. A 2D slice of the 5D likelihood function is shown in Figure 2, which is already non-concave despite the small tree size and projection.

Despite all these challenges, optimization of the likelihood function has proven remarkably effective in practice. For example, Guindon & Gascuel (2003); Guindon et al. (2010) developed PHYML, a coordinate-ascent algorithm that performs well empirically, even with a small number of coordinate updates. Other popular likelihood-based methods include RAxML (Stamatakis, 2014) and IQ-TREE (Nguyen et al., 2015). However, until today, it remains unclear as to why this method works well and there has not been very much theoretical understanding of the optimization landscape of the MLE.

Recent work in statistical estimation theory has highlighted the importance of analyzing the geometric structure of likelihood landscapes (see, e.g., (Ma et al., 2020; Chen & Chen, 2019; Chi et al., 2019)). Very recently, Clancy, Jr. et al. (2025a) showed that the likelihood landscape of the branch length estimation problem in the population limit is very well-conditioned in a box-neighborhood of the true parameter of size $O(1)$. Their population likelihood landscape result is an important building block of the present work on the empirical likelihood landscape and optimization for finding the MLE.

The paper (Clancy, Jr. et al., 2025a) provides bounds for the eigenvalues of the population Hessian in an $L^\infty$-neighborhood of the true parameter $\boldsymbol{\theta}^*$ which is just Step 1 of our three-step program. There are challenges in establishing Steps 2 and 3. Using the population Hessian, it is easy to obtain high probability statements for the empirical Hessian for a fixed parameter $\boldsymbol{\theta}$; however, to do coordinate update

methods we need high probability statements for the Hessian *uniformly* around the true parameter. Uniform matrix concentration requires us to obtain reasonable bounds on the derivative of the Hessian (as a function of the parameter $\boldsymbol{\theta}$) not covered in (Clancy, Jr. et al., 2025a). This implies that the empirical MLE is close to the true parameter with large probability. Furthermore, we analyze a coordinate update algorithm (see Alg. 1) to show that a commonly used optimization algorithm converges to the MLE exponentially fast, provided a sufficiently close initial point.

## 2. The Branch-length Estimation Problem

### 2.1. Problem Statement

Consider a tree $T = (V, E)$ where all nodes have either degree 1 or 3. Information is transmitted from an internal root $\rho$ to the leaves where each edge can corrupt the signal. More precisely, the root node $\rho$ is assigned a uniform spin $\sigma_\rho \in \{\pm 1\}$, this spin is propagated to the leaves where, independently, across each edge $e = \{u, v\}$ the signal is changed $\sigma_u \neq \sigma_v$ with some probability $p_e = \mathbb{P}_{\boldsymbol{\theta}}(\sigma_u \neq \sigma_v)$. Equivalently, $\sigma$ evolves along the edges according to the symmetric two-state inhomogeneous Markov chain with transition matrices $(P_e; e \in E)$

$$P_e = \begin{bmatrix} 1 - p_e & p_e \\ p_e & 1 - p_e \end{bmatrix} = \begin{bmatrix} \frac{1+\theta_e}{2} & \frac{1-\theta_e}{2} \\ \frac{1-\theta_e}{2} & \frac{1+\theta_e}{2} \end{bmatrix}.$$

Here $\boldsymbol{\theta} = (\theta_e; e \in E)$ is a convenient reparametrization of the parameter space. While the possible values for $\theta_e \in [-1, 1]$, we will focus on the "ferromagnetic regime" $\boldsymbol{\theta} \in [0, 1]^E$ where spins at neighboring vertices are positively correlated. That is, it is more likely that there is no signal corruption than there is corruption across each edge.

The problem we are interested in is to recover the true unknown parameter $\boldsymbol{\theta}^*$ from repeated, independent observations of the signals at the leaves of the tree. Namely, let $L$ denote the set of all leaves in the unrooted binary tree $T$ and suppose we have $m$ independent samples $\sigma^{(1)}, \ldots, \sigma^{(m)}$ from the process as described above sampled from the true model $\mathbb{P}_{\boldsymbol{\theta}^*}$. We only get to observe the signals at the leaves, which we denote as $\sigma^{(j)}|_L = (\sigma_v^{(j)}; v \in L)$ for $j = 1, \ldots, m$. The goal is to estimate the true parameter $\boldsymbol{\theta}^*$ from these leaf observations $\sigma^{(1)}, \ldots, \sigma^{(m)}$. This is equivalent to the classical *branch-length estimation problem* (Felsenstein, 1981), as one can view the true parameter $\theta_e^*$ as a monotonic function of the 'length' $l_e^*$ of the edge $e$ along which there is a constant rate (in continuous time) of mutation.

With leaf observations $\sigma^{(1)}|_L, \ldots, \sigma^{(m)}|_L$, we seek to find the maximum likelihood estimator (MLE) as

$$\hat{\boldsymbol{\theta}}_{\text{MLE}} \in \underset{\hat{\boldsymbol{\theta}} \in [-1, 1]^E}{\arg \max} \ \ell(\hat{\boldsymbol{\theta}}; \sigma^{(1)}|_L, \cdots, \sigma^{(m)}|_L). \quad (2)$$

To simplify the notation, throughout we drop the explicit reference to the leaves in all likelihoods.

Here we consider the phylogenetic tree $T$ to be fixed and known so that the MLE problem above is only optimizing over the edge-length parameter $\hat{\theta}$ and not the tree $T$. In general, one is also interested in optimizing over the phylogenetic tree $T$ in the MLE (2); doing so introduces further complications of a combinatorial nature. In particular, this more general problem is known to be NP-hard in the worst case (Chor & Tuller, 2006; Roch, 2006). The work of (Roch & Sly, 2017) (together with the results of (Daskalakis et al., 2011)) shows that an ad hoc polynomial-time algorithm can compute the true parameters with high probability whenever the edge lengths take values in a lattice $\varepsilon \mathbb{Z}^E$, under optimal sample complexity. This present article can be seen as a step towards understanding the properties of optimization approaches, specifically deep inside the Kesten-Stigum reconstruction regime (and without the additional lattice constraint). Many other algorithms with theoretical guarantees are known for reconstructing phylogenetic trees and their branch lengths (see, e.g., (Steel, 2016; Warnow, 2018) and references therein).

# 3. Statement of Results

We formally state our main results in this section. We also sketch the proofs. Detailed proofs are in the appendix. We begin with some assumptions.

## 3.1. Assumptions

Our analysis operates under the assumption that we are well within the Kesten-Stigum reconstruction regime (Kesten & Stigum, 1966; Bleher et al., 1995; Ioffe, 1996), that is, roughly speaking that mutation probabilities are sufficiently small that an ancestral state can be reconstructed with better-than-random accuracy.

Throughout the paper, constants are numbered by the equation in which they appear. We introduce the following restricted parameter spaces that depend on $\delta$.

**Definition 3.1** (Restricted parameter spaces)**.** Let $C_4 > C_3 > c_3 > c_4 > 0$ fixed constants. For a fixed $\delta > 0$, define two subsets $\boldsymbol{\Theta}_0(\delta) \subseteq \widehat{\boldsymbol{\Theta}}_0(\delta) \subset [-1, 1]^E$ by

$$\boldsymbol{\Theta}_0(\delta) := \left\{ (\theta_e = 1 - 2p_e)_{e \in E} \,\middle|\, c_3\delta \leq p_e \leq C_3\delta \;\forall e \in E \right\}$$
$$= [1 - 2C_3\delta, 1 - 2c_3\delta]^E, \tag{3}$$

$$\widehat{\boldsymbol{\Theta}}_0(\delta) := \left\{ (\hat{\theta}_e = 1 - 2\hat{p}_e)_{e \in E} \,\middle|\, c_4\delta \leq \hat{p}_e \leq C_4\delta \;\forall e \in E \right\}$$
$$= [1 - 2C_4\delta, 1 - 2c_4\delta]^E. \tag{4}$$

**A1** (Parameter regime)**.** Assume that $\boldsymbol{\theta}^* \in \boldsymbol{\Theta}_0(\delta)$ and $\hat{\boldsymbol{\theta}} \in \widehat{\boldsymbol{\Theta}}_0(\delta)$. Moreover, the constants $C_4 > C_3 > c_3 > c_4$ satisfy $C_4 \geq 2c_4$.

We will frequently say $\boldsymbol{\theta}^*$ or $\hat{\boldsymbol{\theta}}$ satisfy (3) and (4), respectively, which means that these parameters belong to the sets defined in these equations.

## 3.2. Comments on Notation

Before continuing, we introduce the following convention that is used throughout the article. Given any non-negative function $f(\sigma, \hat{\boldsymbol{\theta}})$ depending on the signals $\sigma = (\sigma_u; u \in T)$ and the estimator $\hat{\boldsymbol{\theta}} \in \widehat{\boldsymbol{\Theta}}_0$ satisfying Assumption **A1**, we write $\mathbb{E}_{\boldsymbol{\theta}^*}[f(\sigma, \hat{\boldsymbol{\theta}})] = O(\delta^\alpha)$ for some $\alpha \in \mathbb{R}$ to mean the there exists a $K(c_3, C_3, c_4, C_4) \in (0, \infty)$ and $\delta_0 = \delta_0(c_3, C_3, c_4, C_4)$ such that

$$\sup_{(\boldsymbol{\theta}, \hat{\boldsymbol{\theta}}) \in \boldsymbol{\Theta}_0 \times \widehat{\boldsymbol{\Theta}}_0} \mathbb{E}_{\boldsymbol{\theta}^*} \left[ f\left(\sigma, \hat{\boldsymbol{\theta}}\right) \right] \leq K\delta^\alpha$$

The constants $K = K(c_3, C_3, c_4, C_4) \in (0, \infty)$ and $\delta_0 = \delta_0(c_3, C_3, c_4, C_4)$ depend on the constants appearing in Assumption **A1**, but otherwise *independent of both the size and topology of the tree $T$*.

We will similarly write $f(\sigma, \hat{\boldsymbol{\theta}}) = O(\delta^\alpha)$ if there exists a constant $K = K(c_3, C_3, c_4, C_4) \in (0, \infty)$ and $\delta_0 = \delta_0(c_3, C_3, c_4, C_4)$ such that for all $\delta \in (0, \delta_0]$ and $(\boldsymbol{\theta}, \hat{\boldsymbol{\theta}}) \in \boldsymbol{\Theta}_0 \times \widehat{\boldsymbol{\Theta}}_0$

$$\mathbb{P}_{\boldsymbol{\theta}^*} \left( f(\sigma, \hat{\boldsymbol{\theta}}) \leq K\delta^\alpha \right) = 1.$$

Here we remind the reader that $\boldsymbol{\Theta}_0, \widehat{\boldsymbol{\Theta}}_0$ are sets that depend on $\delta$. We similarly use $\Omega(\delta^\alpha)$ for analogous statements involving the lower bounds of $K\delta^\alpha$ and we use $\Theta(\delta^\alpha)$ for the functions that are both $O(\delta^\alpha)$ and $\Omega(\delta^\alpha)$.

## 3.3. Empirical Maximum Likelihood Landscape

Our first main result concerns a high-probability characterization of the landscape of the empirical log-likelihood function in (1). Namely, we establish that $\ell$ is $\Theta(\delta^{-1})$-strongly concave and $\Theta(\delta^{-1})$-smooth with probability $1 - \varepsilon$ inside a $L_\infty$-box with universal (tree-independent) diameter $\Theta(\delta)$ around the true parameter $\boldsymbol{\theta}^*$ provided the number of samples $m$ is large enough.

**Theorem 3.2** (Finite-sample log-likelihood landscape: strong concavity and smoothness)**.** *Let $\delta$ be smaller than some universal constant and let $\widehat{\mathbf{H}}$ denote the Hessian of the $m$-sample log-likelihood function $\ell$ in (1). Fix $\varepsilon \in (0, 1)$. Then there exists constants $C_5 > 0$, $\widetilde{C}_6 > C_6 > 0$ s.t. if*

$$m \geq (C_5/\delta)^{\mathrm{diam}(T)+8} \log(\varepsilon^{-1}), \tag{5}$$

*then for any binary tree $T$, and $\boldsymbol{\theta}^* \in \boldsymbol{\Theta}_0(\delta)$,*

$$
\mathbb{P}_{\boldsymbol{\theta}^*}\left( -\widetilde{C}_6\delta^{-1} \leq \inf_{\boldsymbol{\theta}\in\widehat{\boldsymbol{\Theta}}_0(\delta)} \lambda_{\min}(\widehat{\mathbf{H}}(\boldsymbol{\theta})) \right. \tag{6}
$$

$$
\left. \leq \sup_{\boldsymbol{\theta}\in\widehat{\boldsymbol{\Theta}}_0(\delta)} \lambda_{\max}(\widehat{\mathbf{H}}(\boldsymbol{\theta})) \leq -C_6\delta^{-1} \right) \geq 1 - \varepsilon.
$$

The required sample complexity for Theorem 3.2 grows exponentially in the diameter of the tree $T$, which can be as small as $O(\log n)$ when the tree is 'well-balanced'. In such case, Theorem 3.2 below establishes a polynomial sample complexity to obtain a strongly concave and smooth optimization landscape for the MLE problem with high probability. To the best of our knowledge, this is the first result in the literature establishing strong regularity properties of the maximum likelihood landscape for the branch length optimization problem with polynomial sample complexity.

To establish such a result, we use a uniform version of matrix Bernstein's inequality (Lemma 4.5) to show that the Hessian of the empirical log-likelihood function is concentrated near its expectation *uniformly* over the box $\widehat{\boldsymbol{\Theta}}_0(\delta)$ with high probability. Then the assertion will follow from the population landscape result in Lemma 4.4 below.

### 3.4. Statistical and Computational Guarantees

Hereafter, we will denote a generic global maximizer of the empirical log-likelihood function $\ell(\cdot)$ (in (1)) over $\widehat{\boldsymbol{\Theta}}_0(\delta)$ (which always exists) by $\hat{\boldsymbol{\theta}}^*$. A particular consequence of Theorem 3.2 is that $\hat{\boldsymbol{\theta}}^*$ is uniquely determined with high probability and enough samples. In Theorem 3.3 below, we prove a stronger result that with high probability, in addition to the nice geometry of $\ell(\cdot)$ on the box, the MLE $\hat{\boldsymbol{\theta}}^*$ is a $1/\sqrt{m}$-consistent estimator of the true parameter $\boldsymbol{\theta}^*$ with high probability. This consistency (up to a constant depending on $T$) does not depend on our choice of norm on $\mathbb{R}^E$ for any fixed tree. We write $\|\cdot\|$ for the $L^2$-norm of a vector.

**Theorem 3.3** (Statistical estimation guarantee)**.** *Assume the hypothesis of Theorem 3.2 holds. Let $\mathcal{E}_6$ denote the event in (6). Fix $\varepsilon \in (0,1)$. Then there exists a constant $C_7 > 0$ such that, provided $m = \Omega(|E|^2/\varepsilon)$,*

$$
\mathbb{P}_{\boldsymbol{\theta}^*}\left( \mathcal{E}_6 \cap \left\{ \|\boldsymbol{\theta}^* - \hat{\boldsymbol{\theta}}^*\| \leq C_7\sqrt{|E|/m}\log(|E|/\varepsilon) \right\} \right)
$$
$$
\geq 1 - 3\varepsilon. \tag{7}
$$

Now that we know the MLE $\hat{\boldsymbol{\theta}}^*$ is close to the true parameter $\boldsymbol{\theta}^*$ with high probability, we turn our attention to how we can compute the MLE $\hat{\boldsymbol{\theta}}^*$ from the observed samples $\sigma^{(1)}, \ldots, \sigma^{(m)}$ restricted to the leaves. While the log-likelihood function $\ell$ in (1) is non-concave, it has the

nice structure of being strictly concave when restricted to a single branch length $\hat{\theta}_e$ for $e \in E$ (see Lem. 4.2). Thus, it is natural to cycle through the branch lengths and optimize one at a time, maximizing the one-dimensional restricted likelihood function. This yields the following "cyclic coordinate maximization" algorithm for computing the MLE $\boldsymbol{\theta}^*$. Namely, given our estimate $\hat{\boldsymbol{\theta}}_k = (\hat{\theta}_{k;e}; e \in E)$ after $k$ iterations, our algorithm proceeds by optimizing for a single branch length $\theta_{k;e}$ by

$$
\hat{\theta}_{k+1;e} \leftarrow \arg\max_{\hat{\theta}\in[-1,1]} \overline{f}_{k;e}(\hat{\theta}), \quad \text{where} \tag{8}
$$
$$
\overline{f}_{k;e}(\hat{\theta}) := \frac{1}{m}\sum_{i=1}^{m} \ell(\hat{\boldsymbol{\theta}}_{k+1;1:e-1}, \hat{\theta}, \hat{\boldsymbol{\theta}}_{k;e+1:|E|}; \sigma^{(i)}),
$$
$$
\hat{\boldsymbol{\theta}}_{k;i:j} := (\hat{\theta}_{k;i}, \hat{\theta}_{k;i+1}, \cdots, \hat{\theta}_{k;j})
$$

assuming that we label the edge set $E$ as integers from 1 through $|E|$. The one-dimensional objectives $\overline{f}_{k;e}(\hat{\theta})$ in (8) are known to be strictly concave (Fukami & Tateno, 1989) and they have a unique maximizer in $(-1,1)$ at a unique critical point:

$$
\frac{\partial}{\partial\hat{\theta}_e}\overline{f}_{k;e}(\hat{\theta}) = 0. \tag{9}
$$

The unique zero of the above critical-point equation can be found rapidly by using standard zero-finding algorithms (e.g., (Brent, 2013)). See Alg. 1 in Appendix A for a detailed implementation of the algorithm. See e.g. (Guindon & Gascuel, 2003) for a practical implementation of this type of algorithm.

Despite the popularity and the empirical success of the coordinate maximization algorithm above, however, due to the non-concavity of the optimization landscape, there has been no guarantee about the convergence of this algorithm to the maximizer of $\ell$ or the true parameter $\boldsymbol{\theta}^*$. In Theorem 3.4 below, we establish that the coordinate maximization algorithm above (8) converges exponentially fast to the MLE $\hat{\boldsymbol{\theta}}^*$, which is within $C(T,\varepsilon,\delta)m^{-1/2}$ from the true parameter $\boldsymbol{\theta}^*$, provided the initial estimate $\hat{\boldsymbol{\theta}}_0$ is within $O(\delta)$ from the true parameter $\boldsymbol{\theta}$ in $L_2$ norm.

**Theorem 3.4** (Statistical and computational estimation guarantee)**.** *Suppose the hypothesis of Theorem 3.3 holds. Let $(\hat{\boldsymbol{\theta}}_k)_{k\geq 0}$ denote the sequence of estimated parameters generated by the coordinate maximization algorithm (see Alg. 1) with the initial estimate $\hat{\boldsymbol{\theta}}_0$ with $\|\hat{\boldsymbol{\theta}}_0 - \boldsymbol{\theta}^*\| = O(\delta)$. Then with probability at least $1 - 3\varepsilon$, for all $k \geq 0$,*

$$
\|\hat{\boldsymbol{\theta}}^* - \hat{\boldsymbol{\theta}}_k\|^2 \leq \frac{\widetilde{C}_6}{C_6}\left(1 - \frac{C_6}{\widetilde{C}_6}\right)^{k-1}\|\hat{\boldsymbol{\theta}}^* - \hat{\boldsymbol{\theta}}_0\|^2. \tag{10}
$$

*In particular,*

$$\|\boldsymbol{\theta}^* - \hat{\boldsymbol{\theta}}_k\| \leq \underbrace{C_7 \sqrt{|E|/m} \log(|E|/\varepsilon)}_{=\text{statistical error}} \tag{11}$$

$$+ \underbrace{\sqrt{\frac{\widetilde{C_6}}{C_6}} \left(1 - \frac{C_6}{\widetilde{C_6}}\right)^{(k-1)/2} \|\hat{\boldsymbol{\theta}}^* - \hat{\boldsymbol{\theta}}_0\|}_{=\text{computational error}}.$$

It is important to note that the exponential rate of convergence of the coordinate maximization in Theorem 3.4, $\frac{\widetilde{C_6}}{C_6} \in (0, 1)$, is a universal constant that does not depend on the tree $T$ and also the parameter $\delta$ (as long as it is less than some universal constant in Thm. 3.2). This means that the computational error for computing the MLE can be made to be less than a desired tolerance $\varepsilon$ within $C \log \varepsilon^{-1}$ iterations for some universal constant $C$. This gives some theoretical support for the empirical fact that coordinate maximization algorithm performs well empirically, even with a small number of coordinate updates (Guindon & Gascuel, 2003; Guindon et al., 2010).

It would be of interest to show that the assumption that $\|\hat{\boldsymbol{\theta}}_0 - \boldsymbol{\theta}^*\| = O(\delta)$ can be dropped; however, our proof needs the initial iterate to be sufficiently close to $\boldsymbol{\theta}^*$ in order to know that the empirical Hessian is smooth and strongly concave with high probability (Thm. 3.2) and that the subsequent iterates also lie in this "good" region.

## 4. Sketch of Proofs

### 4.1. Characterizing the Likelihood Landscape Using Magnetization

Fix two distinct nodes $u, v$ in $T$. We call a node $w$ a *descendant* of $u$ with respect to node $v$ if the shortest path between $w$ and $v$ contains $u$. The *descendant subtree at $u$ with respect to $v$* is the subtree $T_u$ rooted at $u$ consisting of all descendants of $u$ with respect to $v$. A subtree of $T$ rooted at $u$ is a *descendant subtree of $u$* if it is a descendant subtree of $u$ with respect to some node $v$.

The following notion of 'magnetization' is central to computing the derivatives of the log-likelihood. Roughly speaking, the magnetization $Z_u$ of a node $u$ with respect to a descendant subtree $T_u$ rooted at $u$ is the 'bias' on its spin after observing all spins at the leaves of the descendant subtree $T_u$. For instance, if all spins on the leaves of $T_u$ are $+$, then $u$ will be quite likely to have $+$ spin as well. The formal definition of magnetization is given below.

**Definition 4.1** (Magnetization). Let $T_u$ be a descendant subtree of $T$ rooted at a node $u$. Let $L_u$ denote the set of all leaves in $T_u$. For a generic parameter $\hat{\boldsymbol{\theta}} \in [0, 1]^{E(T_u)}$ and spin configuration $\tau \in \{\pm\}^{L_u}$ on the leaves of $T_u$, define

the magnetization at the root $u$ of $T_u$ as $Z_u = Z_u(\hat{\boldsymbol{\theta}}; \sigma)$ by

$$Z_u = \mathbb{P}_{\hat{\boldsymbol{\theta}}}(\hat{\sigma}_u = +1 \mid \hat{\sigma}_{L_u} = \sigma_{L_u}) \tag{12}$$
$$- \mathbb{P}_{\hat{\boldsymbol{\theta}}}(\hat{\sigma}_u = -1 \mid \hat{\sigma}_{L_u} = \sigma_{L_u}),$$

where $\hat{\sigma}$ is a random spin configuration on $T$ sampled from $\mathbb{P}_{\hat{\boldsymbol{\theta}}}$.

If $T_u$ consists of a single node $u$, then $Z_u = \sigma_u$ as we get to observe the spin at $u$. In general, $Z_u$ is a random variable determined by the spin configuration $\sigma_{L_u}$ on the leaves of $T_u$ and takes values in $[-1, 1]$. The magnetization $Z_u$ at a node $u$ also depends implicitly on the choice of the descendant subtree $T_u$. In (Borgs et al., 2006), Borgs et al. established that the magnetization of a root on any tree can be obtained as an explicit function of the magnetizations of the descendant subtrees for all the children and the edge parameters. We recall this in the appendix, see (29) in particular.

Consider the log-likelihood function $\ell(\hat{\boldsymbol{\theta}}; \sigma)$ in (1) and two edges $e = \{x, y\}$ and $f = \{u, v\} \in E(T)$. Let $T_x$ and $T_y$ denote the subtrees rooted at $x$ and $y$ (resp.) obtained by removing $e$ from the edges of $T$. Suppose that $f \in E(T_y)$ and that $u$ is closer to $y$ than $v$, i.e., $d(u, y) < d(v, y)$. Enumerate the vertices on the path from $y$ to $u$ by $y = y_N, y_{N-1}, \ldots, y_1, y_0 = u$ and set $y_{-1} = v$ and $y_{N+1} = x$. Note that for each vertex $y_j$ with $j \in \{0, \ldots, N\}$, the vertex $y_j$ has degree three and so has neighbors $\{y_{j-1}, y_{j+1}, w_j\}$ for some other vertex $w_j$. Accordingly, we have $T_x = T_{y_{N+1}}$ and $T_y = T_{y_N}$, and for every node $z$ in $T_y$, the descendant subtree $T_z$ is with respect to the root $x$.

The following key lemma relates the derivatives of the log-likelihood and the magnetizations.

**Lemma 4.2** (Likelihood and magnetization). *The following formulas hold.*

**(i)** *(Gradient) For edge $e = \{x, y\}$, we have*

$$\frac{\partial}{\partial \hat{\theta}_e} \ell(\hat{\boldsymbol{\theta}}; \sigma) = \frac{Z_x Z_y}{1 + Z_x Z_y \hat{\theta}_e}, \tag{13}$$

**(ii)** *(Hessian) For edges $e = \{x, y\}$ and $f = \{u, v\}$ with* $\text{dist}(e, f) = N$ *as above, we have*

$$\frac{\partial^2}{\partial \hat{\theta}_e \partial \hat{\theta}_f} \ell(\hat{\boldsymbol{\theta}}; \sigma) = \left( \frac{\hat{\theta}_e Z_x Z_v}{(1 + \hat{\theta}_e Z_x Z_y)^2} \prod_{j=1}^{N} \hat{\theta}_{\{y_j, y_{j-1}\}} \right)$$

$$\times \prod_{j=0}^{N} \frac{\left(1 - (\hat{\theta}_{\{y_j, w_j\}} Z_{w_j})^2\right)}{\left(1 + \hat{\theta}_{\{y_j, w_j\}} \hat{\theta}_{\{y_j, y_{j-1}\}} Z_{w_j} Z_{y_{j-1}}\right)^2}. \tag{14}$$

**(iii)** *(Third-order derivatives) If $\hat{\boldsymbol{\theta}} \in \widehat{\boldsymbol{\Theta}}_0(\delta)$, then for all edges $e_1, e_2, e_3$:*

$$\left| \frac{\partial^3}{\partial \hat{\theta}_{e_1} \partial \hat{\theta}_{e_2} \partial \hat{\theta}_{e_3}} \ell(\hat{\boldsymbol{\theta}}; \sigma) \right| \leq \frac{4 \operatorname{diam}(T)}{(2c_4 \delta)^{4 \operatorname{diam}(T) + 2}}. \tag{15}$$

The expression for the Hessian in (14) above is rather complicated. Looking at the denominators in (14), we see that each of them is (at worst) $\Omega(\delta^2)$, and as there are most $\mathrm{diam}(T)$ many a naïve bound on the Hessain gives

$$\left| \frac{\partial^2}{\partial \hat{\theta}_e \partial \hat{\theta}_f} \ell(\hat{\boldsymbol{\theta}}; \sigma) \right| = O\left( \delta^{-2\,\mathrm{diam}(T)} \right);$$

however, we can provide a much better bound. We state this as the following lemma.

**Lemma 4.3.** *There exists constants $C_{16}$, $\widetilde{C}_{16}$ and $\delta_{16}$ such that for all binary trees $T$ and $\delta \leq \delta_{16}$*

$$\left| \frac{\partial^2}{\partial \hat{\theta}_e \partial \hat{\theta}_f} \ell(\hat{\boldsymbol{\theta}}; \sigma) \right| \leq C_{16} \left( \frac{\widetilde{C}_{16}}{\delta} \right)^{\mathrm{diam}(T)/2+4}. \quad (16)$$

### 4.2. Sketch of the Proof of Thm. 3.2

To analyze the MLE landscape in (1) we rely on knowledge of the population landscape obtained in (Clancy, Jr. et al., 2025a) which we now recall. Define $\mathbf{H}(\hat{\boldsymbol{\theta}})$ to be the Hessian of the one-sample population log-likelihood

$$\mathbf{H}(\hat{\boldsymbol{\theta}}) = D^2 \mathbb{E}_{\boldsymbol{\theta}^*}\left[ \ell(\hat{\boldsymbol{\theta}}; \sigma) \right] = \mathbb{E}_{\boldsymbol{\theta}^*}\left[ D^2 \ell(\hat{\boldsymbol{\theta}}; \sigma) \right] \quad (17)$$

$$= \left( \mathbb{E}_{\boldsymbol{\theta}^*}\left[ \frac{\partial^2}{\partial \hat{\theta}_e \partial \hat{\theta}_f} \ell(\hat{\boldsymbol{\theta}}; \sigma) \right]; e, f \in E(T) \right). \quad (18)$$

In (Clancy, Jr. et al., 2025a), the following population likelihood landscape result is shown:

**Theorem 4.4** (Population log-likelihood landscape: strong concavity and smoothness)**.** *There exists a constant $\delta_{19} \in (0,1)$ depending only on $c_3, C_3, c_4, C_4$ such that for all binary trees $T$, $\delta \leq \delta_{19}$ and $\hat{\boldsymbol{\theta}} \in \widehat{\boldsymbol{\Theta}}_0(\delta)$ and $\boldsymbol{\theta}^* \in \boldsymbol{\Theta}_0(\delta)$*

$$-\frac{\widetilde{C}_{19}}{\delta} \leq \mathbb{E}_{\boldsymbol{\theta}}\left[ \frac{\partial^2}{\partial \hat{\theta}_e^2} \ell(\hat{\boldsymbol{\theta}}; \sigma) \right] \leq -\frac{C_{19}}{\delta} \quad \text{for all } e \in E(T)$$

$$-\frac{\widetilde{C}_{19}}{\delta} - 26 \leq \lambda_{\min}(\mathbf{H}(\hat{\boldsymbol{\theta}})) \leq \lambda_{\max}(\mathbf{H}(\hat{\boldsymbol{\theta}})) \leq -\frac{C_{19}}{\delta} + 26, \quad (19)$$

*where $\lambda_{\min}(\cdot)$ and $\lambda_{\max}(\cdot)$ denote the minimum and the maximum eigenvalues of a matrix. In particular, in the population limit $m \to \infty$, the log-likelihood function $\ell$ in (2) is $(\frac{C_{19}}{\delta} - 26)$ - strongly concave and $(\frac{\widetilde{C}_{19}}{\delta} + 26)$-smooth. In particular, the true parameter $\boldsymbol{\theta}^*$ is the unique maximizer of $\ell$ over $\widehat{\boldsymbol{\Theta}}_0$.*

Now, how do we transfer this population landscape result to the empirical landscape with high probability? We essentially need some type of uniform concentration bound on parameterized matrix-valued random functions around its expectation. To this end, we establish a uniform version of the well-known matrix Bernstein inequality (Thm. 1.4 in (Tropp, 2012)) for sums of self-adjoint independent random matrices. Given a pre-compact parameter space $\Theta \subset \mathbb{R}^p$, we let $\|\Theta\| := \sup_{x,y \in \Theta} \|x - y\|$ denote the ($L^2$-)diameter of $\Theta$.

**Lemma 4.5** (Uniform Matrix Bernstein)**.** *Fix a compact parameter space $\Theta \subseteq \mathbb{R}^p$. Consider a finite sequence $(X_k(\theta))_{1 \leq k \leq n}$ of self-adjoint independent random matrices in dimension $d$ parameterized by $\theta \in \Theta$. Assume that there exists a constant $R \geq 0$ such that for each $\theta \in \Theta$, $1 \leq k \leq n$ that almost surely*

$$\mathbb{E}[X_k(\theta)] = O \quad \text{and} \quad \lambda_{\max}(X_k(\theta)) \leq R$$

*Furthermore, suppose that the random matrices depend on the parameter smoothly: There exists a constant $L > 0$ such that*

$$\|X_k(\theta) - X_k(\theta')\|_2 \leq L\|\theta - \theta'\| \quad (20)$$

*almost surely for all $1 \leq k \leq n$ and $\theta, \theta' \in \Theta$. Then for all $t \geq 0$, denoting $\sigma^2 := \sup_{\theta \in \Theta} \left\| \sum_k \mathbb{E}\left[ X_k(\theta)^2 \right] \right\|_2$,*

$$\mathbb{P}\left( \sup_{\theta \in \Theta} \left\| \sum_k X_k(\theta) \right\|_2 \geq t \right) \quad (21)$$

$$\leq 2d \|\Theta\|^p \left( 1 + \frac{4nL}{t} \right)^p \exp\left( \frac{-t^2/8}{\sigma^2 + Rt/6} \right).$$

*Sketch of proof.* The statement can be deduced by using an $\varepsilon$-net argument with the standard matrix Bernstein inequality. Lipschitz continuity of the parameterized random matrix $X_k(\cdot)$ is needed to do so. See Appendix B for more details. $\square$

Now we sketch the proof of Theorem 3.2. The key is to look at the fluctuation of the empirical Hessian about its population expectation:

$$\widehat{\mathbf{H}}(\boldsymbol{\theta}) - \mathbf{H}(\boldsymbol{\theta}) = \sum_{k=1}^{m} \underbrace{m^{-1}(\mathbf{H}_k(\boldsymbol{\theta}) - \mathbf{H}(\boldsymbol{\theta}))}_{=:X_k(\boldsymbol{\theta})}, \quad (22)$$

where $\mathbf{H}_k$ denotes the (random) Hessian of the $k$th leaf observation $\sigma^{(i)}$. Once we verify the hypothesis of Lemma 4.5, it implies that the random matrix on the left-hand side above has a small spectral norm. By Weyl's inequality, this implies that the maximum eigenvalue of the empirical Hessian is concentrated around that of the expected Hessian: Uniformly over all $\boldsymbol{\theta} \in \widehat{\boldsymbol{\Theta}}_0(\delta)$,

$$|\lambda_{\max}(\widehat{\mathbf{H}}(\boldsymbol{\theta})) - \lambda_{\max}(\mathbf{H}(\boldsymbol{\theta}))| \leq \|\widehat{\mathbf{H}}(\boldsymbol{\theta}) - \mathbf{H}(\boldsymbol{\theta})\|_2 \leq 1$$

with high probability (provided $m$ is sufficiently large).

In order to verify the hypothesis of the uniform matrix Bernstein inequality (Lem. 4.5), we crucially use the deterministic bounds on the entries of the Hessian (in Lem. 4.3) and the third-order derivatives of the log-likelihood (in Lem. 4.2). Namely, by Lem. 4.3, the entries in the Hessian $\mathbf{H}_k(\boldsymbol{\theta})$ at any parameter $\boldsymbol{\theta} \in \widehat{\boldsymbol{\Theta}}_0(\delta)$ are uniformly bounded by $J = \delta^{-O(\mathrm{diam}(T)/2)}$. Hence the entries of the deviation matrix $X_k(\boldsymbol{\theta})$ are also uniformly bounded by $J$. Then by Gershgorin's circle theorem, it follows that

$$\|X_k(\boldsymbol{\theta})\|_2 \leq |E|J = |E|\delta^{-O(\mathrm{diam}(T)/2)} \qquad (23)$$

uniformly over all $\boldsymbol{\theta} \in \widehat{\boldsymbol{\Theta}}_0(\delta)$, as this is an upper bound of the absolute row sums.

Next, note that each entry of $\mathbf{H}_k(\boldsymbol{\theta})^2$ is uniformly bounded by $|E|J^2$, so by a similar argument and an application of Weyl's inequality, the maximum eigenvalue of

$$\sigma^2 \leq \sum_{k=1}^m \left\| \mathbb{E}[X_k(\boldsymbol{\theta})^2] \right\|_2 = O(m|E|^2 J^2). \qquad (24)$$

Lastly, for the Lipschitz continuity in (20), we may use the mean value theorem and the uniform bound on the third-order derivative in Lem. 4.2 to deduce

$$\begin{aligned}
\|\mathbf{H}_k(\boldsymbol{\theta}) &- \mathbf{H}_k(\boldsymbol{\theta}')\|_2 \\
&\leq |E|^2 \|\mathbf{H}_i(\boldsymbol{\theta}) - \mathbf{H}_i(\boldsymbol{\theta}')\|_{\max} \qquad (25) \\
&\leq M|E|^2 \|\boldsymbol{\theta} - \boldsymbol{\theta}'\| \qquad (26) \\
&= |E|^2 \mathrm{diam}(T)\delta^{-O(\mathrm{diam}(T))}\|\boldsymbol{\theta} - \boldsymbol{\theta}'\|. \qquad (27)
\end{aligned}$$

for all $\boldsymbol{\theta}, \boldsymbol{\theta}' \in \widehat{\boldsymbol{\Theta}}_0(\delta)$, where $M$ denotes the largest absolute value of the third-order derivatives of $\ell(\boldsymbol{\theta}; \sigma^{(i)})$ over all $\boldsymbol{\theta} \in \widehat{\boldsymbol{\Theta}}_0(\delta)$. Since the above bound holds almost surely for all parameters in the box, it also holds for the expected Hessians, and hence for their deviations from the mean.

### 4.3. Sketch of the Proof of Thm. 3.3

The reason why we should expect to see an $O(1/\sqrt{m})$ error term appearing in Theorem 3.3 is the central limit theorem tells us that $\ell(\boldsymbol{\theta})$ should be within $O(m^{-1/2})$ of $\mathbb{E}[\ell(\boldsymbol{\theta})]$ as it has finite variance (Lemma 4.3). We can make this quantitative and non-asymptotic with the Berry-Esseen theorem. In order to turn this idea into a statement about the maximizers, we use a first-order Taylor expansion of their difference at $\boldsymbol{\theta}^*$.

In standard maximum likelihood analysis, one uses the Fisher information (expected Hessian at the true parameter) for the second-order term and controls the error subsumed in the third-order term. But in our context, this requires us to understand the continuity of the third-order derivative of the log-likelihood, which in turn reduces to bounding the

fourth-order derivative of the likelihood function, which is somewhat complicated. Instead, we directly use the random empirical Hessian for the second order term and use our previous result that with high probability, the empirical Hessian has bounded maximum eigenvalue uniformly within a 'good box' in the parameter space.

To sketch the proof of Theorem 3.3, we will write $\ell(\hat{\boldsymbol{\theta}}) = \ell(\hat{\boldsymbol{\theta}}; x_1, \ldots, x_m) = m^{-1} \sum_{j=1}^m \ell(\hat{\boldsymbol{\theta}}; x_j)$. Then, by a first-order Taylor expansion around $\boldsymbol{\theta}^*$

$$\begin{aligned}
\ell(\hat{\boldsymbol{\theta}}) - \ell(\boldsymbol{\theta}^*) &= \langle \nabla_{\hat{\boldsymbol{\theta}}} \ell(\boldsymbol{\theta}^*), \hat{\boldsymbol{\theta}} - \boldsymbol{\theta}^* \rangle + O\left( \|\hat{\boldsymbol{\theta}} - \boldsymbol{\theta}^*\|^2 \right) \\
&= \frac{\|\hat{\boldsymbol{\theta}} - \boldsymbol{\theta}^*\|}{\sqrt{m}} T_m(\hat{\boldsymbol{\theta}}) + O(\|\hat{\boldsymbol{\theta}} - \boldsymbol{\theta}^*\|^2)
\end{aligned}$$

where

$$T_m(\hat{\boldsymbol{\theta}}) = \frac{\sqrt{m}}{\|\hat{\boldsymbol{\theta}} - \boldsymbol{\theta}^*\|} \langle \nabla_{\hat{\boldsymbol{\theta}}} \ell(\boldsymbol{\theta}^*) - \mathbb{E}[\nabla_{\hat{\boldsymbol{\theta}}} \ell(\boldsymbol{\theta}^*)], \hat{\boldsymbol{\theta}} - \boldsymbol{\theta}^* \rangle$$

and the big-$O$ error term depends on the empirical Hessian. This allows us to write the big-$O$ error term above in terms of some (explicit) function of the empirical Hessian of $\ell(\hat{\boldsymbol{\theta}})$, say $\Lambda_m$. In turn, we can say

$$\ell(\hat{\boldsymbol{\theta}}) - \ell(\boldsymbol{\theta}^*) \leq \frac{\|\hat{\boldsymbol{\theta}} - \boldsymbol{\theta}^*\|}{\sqrt{m}} T_m(\hat{\boldsymbol{\theta}}) + \Lambda_m \|\hat{\boldsymbol{\theta}} - \boldsymbol{\theta}^*\|^2.$$

If we restrict our attention to $\hat{\boldsymbol{\theta}}$ such that $\|\hat{\boldsymbol{\theta}} - \boldsymbol{\theta}^*\| = Rm^{-1/2}$ for some $R > 0$

$$\frac{\ell(\hat{\boldsymbol{\theta}}) - \ell(\boldsymbol{\theta}^*)}{\|\boldsymbol{\theta} - \boldsymbol{\theta}^*\|^2} \leq \frac{1}{R} T_m(\hat{\boldsymbol{\theta}}) + \Lambda_m. \qquad (28)$$

If we can show that there is some non-random choice of $R$ (depending on $T$ and $\varepsilon$) such that the right-hand side above is strictly negative then we can use our previous result on the concavity of $\ell(\hat{\boldsymbol{\theta}})$ near $\boldsymbol{\theta}^*$ to conclude that the maximizer satisfies $\|\hat{\boldsymbol{\theta}}^*_{\mathrm{MLE}} - \boldsymbol{\theta}^*\| \leq Rm^{-1/2}$.

This final task boils down to showing that $T_m(\hat{\boldsymbol{\theta}})$ concentrates around 0 uniformly in $\hat{\boldsymbol{\theta}}$ close to $\boldsymbol{\theta}^*$ and showing that there is some $t > 0$ such that $\mathbb{P}_{\boldsymbol{\theta}^*}(\Lambda_m < -t) \geq 1 - \varepsilon$. We can use Berry-Esseen to accomplish the former and our previous result controlling the eigenvalues of the empirical Hessian accomplishes the latter.

### 4.4. Sketch of the Proof of Thm. 3.4

As already mentioned, the concavity of the empirical log-likelihood, showing that the right-hand side of (28) is strictly negative with large probability implies that the (unique) maximizer satisfies $\|\hat{\boldsymbol{\theta}}^*_{\mathrm{MLE}} - \boldsymbol{\theta}^*\| \leq Rm^{-1/2}$. One of the key steps in proving Theorem 3.4 is to show that not only in the global maximizer within $Rm^{-1/2}$ of $\boldsymbol{\theta}^*$ but also all the

iterates from the coordinate maximization algorithm (Alg. 1) are also within $Rm^{-1/2}$ of $\boldsymbol{\theta}^*$. That means we do not escape an $L_2$ neighborhood of $\boldsymbol{\theta}^*$ with large probability whenever we start with a good initialization.

The last step is to analyze the coordinate maximization algorithm without confinement constraints. In the optimization literature, the block coordinate *minimization* iterates for $\rho$-strongly convex and smooth landscapes with two blocks is known to converge linearly (Beck & Tetruashvili, 2013). Moreover, the rate of convergence depends on the parameter $\rho$ and the block-smoothness parameter $L_1, L_2$. We extend this to arbitrarily many blocks to deal with arbitrarily large trees.

**Lemma 4.6** (Block coordinate minimization for strongly convex and smooth objectives). *Fix a parameter space $\Omega := \prod_{i=1}^{b} I_i \subseteq \mathbb{R}^p$, where $I_i \subseteq \mathbb{R}^{p_i}$ is an open convex subset with $p_1 + \cdots + p_b = p$ for some $b \in \{1, \ldots, p\}$. Let $f : \Omega \to \mathbb{R}$ be a $\rho$-strongly convex function for some $\rho > 0$. Further assume that, there exists constants $L_1, \ldots, L_b$ such that $f$ restricted on the $i$th block $I_i$ is $L_i$-smooth for $i = 1, \ldots, b$. Consider the following cyclic block coordinate minimization algorithm: Given $\boldsymbol{\theta}_{n-1} \in \Omega$, $\boldsymbol{\theta}_n = (\theta_{n;i}; i = 1, \ldots, b)$ be as*

$$\theta_{n;i} \leftarrow \underset{\theta \in I_i}{\arg\min} \, f(\theta_{n;1}, \ldots, \theta_{n;i-1}, \theta, \theta_{n-1;i+1}, \ldots, \theta_{n-1;b}).$$

*Assume these iterates are well-defined. Suppose $f^* := \min_{\boldsymbol{\theta} \in \Omega} f(\boldsymbol{\theta}) > -\infty$ and the minimum is attained. Then for all $n \geq 1$,*

$$f(\boldsymbol{\theta}_n) - f^* \leq \left( 1 - \frac{\rho}{\min\{L_1, \ldots, L_b\}} \right)^{n-1} (f(\boldsymbol{\theta}_0) - f^*).$$

## Conclusion

In this work, we analyzed a maximum likelihood estimation (MLE) problem with a non-concave likelihood landscape. Specifically, for branch-length estimation in phylogenetics, we showed that even when the likelihood landscape may contain multiple local maxima, one can identify a semi-global region where the landscape behaves well – exhibiting strong concavity and containing a unique maximizer. Our analysis demonstrates that with polynomial sample complexity when the tree is balanced, the empirical likelihood concentrates around its population counterpart within this region.

Relying strong concavity of the population likelihood in a suitable neighborhood, the key steps of our approach are: proving Lipschitz continuity of the population Hessian entries and bounding the spectral norm of the per-sample empirical Hessian. While we focused on branch-length estimation, some aspects of our methodology may be relevant to a broader class of non-concave maximum likelihood problems where similar "benign" non-concavity structures exists.

## Acknowledgments

HL was partially supported by NSF grant DMS-2206296. DC and SR were partially supported by the Institute for Foundations of Data Science (IFDS) through NSF grant DMS-2023239 (TRIPODS Phase II). It is also based upon work supported by the NSF under grant DMS-1929284 while one of the authors (SR) was in residence at the Institute for Computational and Experimental Research in Mathematics (ICERM) in Providence, RI, during the Theory, Methods, and Applications of Quantitative Phylogenomics semester program. SR was also supported by NSF grant DMS-2308495, as well as a Van Vleck Research Professor Award and a Vilas Distinguished Achievement Professorship.

## Impact statement

This paper presents work whose goal is to advance the field of Machine Learning. There are many potential societal consequences of our work, none of which we feel must be specifically highlighted here.

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

# Sample complexity of branch-length estimation by maximum likelihood
## Supplementary Material

## A. A detailed implementation of coordinate maximization algorithm (8)

Here we provide a detailed implementation of the coordinate maximization algorithm (8) for branch length estimation. A key step is to solve the critical point equation (9), for which we need to compute the first-order derivatives of the log-likelihood function. In (Clancy, Jr. et al., 2025b), Clancy et al. obtained a simple recursive formula for the gradient of the log-likelihood of a single sample using the magnetization in Def. 4.1. In particular, they show for a single sample $\sigma$ that

$$\frac{\partial}{\partial \hat{\theta}_e} \ell(\hat{\boldsymbol{\theta}}; \sigma) = \frac{Z_u(\hat{\boldsymbol{\theta}}; \sigma) Z_v(\hat{\boldsymbol{\theta}}; \sigma)}{1 + \hat{\theta}_e Z_u(\hat{\boldsymbol{\theta}}; \sigma) Z_v(\hat{\boldsymbol{\theta}}; \sigma)}$$

where $e = \{u, v\}$ and $Z_u(\hat{\boldsymbol{\theta}}; \sigma)$, $Z_v(\hat{\boldsymbol{\theta}}; \sigma)$ can computed recursively as follows. Consider the subtree $T_u$ of $T$ obtained by deleting the edge $e$ and rooted at $u$. For all the leafs $x \in T_u$ define $Z_x = Z_x(\hat{\boldsymbol{\theta}}; \sigma) = \sigma_x$ and for any other vertex $x$ with children $a, b$ define

$$Z_x = q(\hat{\boldsymbol{\theta}}_{\{x,a\}} Z_a, \hat{\boldsymbol{\theta}}_{\{x,b\}} Z_b) \tag{29}$$

where $q(s, t) = \frac{s+t}{1+st}$. This yields the following explicit and easily executable implementation of the coordinate maximization algorithm in (8).

---

**Algorithm 1** Empirical likelihood maximization by cyclic coordinate maximization

---

1: **Input:** $T = (V, E)$ (Binary phylogenetic tree); $\hat{\boldsymbol{\theta}}_0 = (\hat{\theta}_{0;e})_{e \in E}$ (initial estimate); $m$ (number of samples); $M$ (number of iterations); $\tau > 0$ (optional step-size)
2: Sample $m$ i.i.d. spin configurations $\sigma^{(1)}, \ldots, \sigma^{(m)}$ on $T$ under the true model $\mathbb{P}_{\boldsymbol{\theta}^*}$
3: **for** $k = 0, \ldots, M$ **do**:
4:     Set $\hat{\boldsymbol{\theta}}_{k+1} = (\hat{\theta}_{k+1;e})_{e \in E} \leftarrow \hat{\boldsymbol{\theta}}_k$
5:     **for** edges $e = uv \in E$ **do**:
6:       **for** $i = 1, \ldots, m$ **do**:
7:         Compute the magnetizations $Z_{k;u}^{(i)}$ and $Z_{k;v}^{(i)}$ using $\sigma^{(i)}|L$ and $\hat{\boldsymbol{\theta}}_{k+1}$
8:       **end for**
9:     Compute the empirical gradient: $\frac{\partial}{\partial \hat{\theta}_e} \bar{f}_{k;e}(\hat{\theta}_e) := \frac{1}{m} \sum_{i=1}^m \frac{Z_{k;u}^{(i)} Z_{k;v}^{(i)}}{1 + \hat{\theta}_e Z_{k;u}^{(i)} Z_{k;v}^{(i)}}$    ($\triangleright$ *a rational function $\hat{\theta}_e$*)
10:     $\hat{\theta}_{k+1;e} \leftarrow$ zero of $\frac{\partial}{\partial \hat{\theta}_e} \bar{f}_{k;e}(\hat{\theta}_e) = 0$ in $[-1, 1]$    ($\triangleright$ *update the coordinate $e$ of $\hat{\boldsymbol{\theta}}_{k+1}$ by coordinate maximization*)
11:     **end for**
12: **end for**
13: **output:** $\hat{\boldsymbol{\theta}}_M$

---

## B. A uniform matrix Bernstein's inequality

*Proof of Lemma 4.5.* Since $\Theta \subseteq \mathbb{R}^p$ is compact, it can be covered by a finite number of $L^2$-balls of any given radius $\varepsilon > 0$. Let $\mathcal{U}_\varepsilon$ denote a smallest collection of $\varepsilon$-balls that cover $\Theta$ and let $N(\varepsilon) := |\mathcal{U}_\varepsilon|$ denote the smallest number of $\varepsilon$-balls to cover $\Theta$. It is easy to verify (see, e.g., (Roch, 2024))

$$N(\varepsilon) \le \|\Theta\|^p (1 + (2/\varepsilon))^p. \tag{30}$$

Let $\theta_1, \cdots, \theta_{N(\varepsilon)}$ be the centers of $\varepsilon$-balls in $\mathcal{U}_\varepsilon$. Then for each $\theta \in \Theta$, there exists $1 \le j \le N(\varepsilon)$ such that $\|\theta - \theta_j\| < \varepsilon$.

Next, denote $Y(x) := \sum_k X_k(x)$ for each $x \in \Theta$, which is self-adjoint. By Weyl's inequality and the hypothesis, for each $x, y \in \Theta$,

$$|\lambda_{\max}(Y(x)) - \lambda_{\max}(Y(y))| \le \|Y(x) - Y(y)\|_2 \le \sum_k \|X_k(x) - X_k(y)\|_2 \le nL\|x - y\|.$$

Hence for each $\theta \in \Theta$, there exists $1 \leq j \leq N(\varepsilon)$ such that

$$
\begin{aligned}
\lambda_{\max}(Y(\theta)) &\leq \lambda_{\max}(Y(\theta_j)) + |\lambda_{\max}(Y(\theta)) - \lambda_{\max}(Y(\theta_j))| \\
&\leq \lambda_{\max}(Y(\theta_j)) + nL\varepsilon.
\end{aligned}
$$

If follows that, by choosing $\varepsilon = \frac{t}{2nL}$ and using a union bound with (30),

$$
\begin{aligned}
\mathbb{P}\left(\sup_{\theta \in \Theta} \lambda_{\max}(Y(\theta)) \geq t\right) &\leq \sum_{j=1}^{N\left(\frac{t}{2nL}\right)} \mathbb{P}\left(\lambda_{\max}(Y(\theta_j)) \geq t/2\right) \\
&\leq d\,\|\Theta\|^p \left(1 + \frac{4nL}{t}\right)^p \exp\left(\frac{-t^2/8}{\sigma^2 + Rt/6}\right).
\end{aligned}
$$

Here the second inequality uses the standard matrix Bernstein inequality for self-adjoint random matrices (see, Thm.1.4 in (Tropp, 2012)), where $\sigma^2$ is defined in the statement.

To finish, note that

$$
\sup_{\theta \in \Theta} \max_{1 \leq i \leq d} |\lambda_i(Y(\theta))| \leq \max\left\{\sup_{\theta \in \Theta} \max_{1 \leq i \leq d} \lambda_i(Y(\theta)), \ \sup_{\theta \in \Theta} \max_{1 \leq i \leq d} \lambda_i(-Y(\theta))\right\}.
$$

Therefore, applying the same argument for $-Y(\theta)$ and combining the resulting concentration bounds using a union bound, we can derive (21). $\qquad\square$

## C. Proof of results for statistical and computational guarantees

In this section, we prove Theorems 3.2, 3.3, and 3.4.

We now prove Theorem 3.2, which we re-state below with more explicit constants.

**Theorem C.1** (Finite-sample log-likelihood landscape: strong concavity and smoothness, Thm. 3.2)**.** *Let $\delta \leq \delta_{19}$ and let $\widehat{\mathbf{H}}$ denote the Hessian of the $m$-sample log-likelihood function $\ell$ in (1). Fix $\varepsilon \in (0,1)$. Then there exists a constant $C_{31} > 0$ such that if*

$$
m \geq (C_{31}/\delta)^{\mathrm{diam}(T)+8} \log(\varepsilon^{-1}), \tag{31}
$$

*then for any binary tree $T$, and $\boldsymbol{\theta}^* \in \boldsymbol{\Theta}_0(\delta)$,*

$$
\mathbb{P}_{\boldsymbol{\theta}^*}\left(-\frac{\widetilde{C}_{19}}{\delta} - 27 \leq \inf_{\boldsymbol{\theta} \in \widehat{\boldsymbol{\Theta}}_0(\delta)} \lambda_{\min}(\widehat{\mathbf{H}}(\boldsymbol{\theta})) \leq \sup_{\boldsymbol{\theta} \in \widehat{\boldsymbol{\Theta}}_0(\delta)} \lambda_{\max}(\widehat{\mathbf{H}}(\boldsymbol{\theta})) \leq -\frac{C_{19}}{\delta} + 27\right) \geq 1 - \varepsilon. \tag{32}
$$

*Proof.* Recall that we denote by $\mathbf{H}(\hat{\boldsymbol{\theta}})$ the Hessian of the population likelihood (see (17)). Let $\mathbf{H}_i(\hat{\boldsymbol{\theta}}) := \nabla_{\hat{\boldsymbol{\theta}}} \nabla_{\hat{\boldsymbol{\theta}}^T} \ell(\hat{\boldsymbol{\theta}}; x_i)$ denote the Hessian of the log-likelihood of the $i$th configuration $x_i$ at parameter $\hat{\boldsymbol{\theta}}$. Then $\widehat{\mathbf{H}}(\hat{\boldsymbol{\theta}}) := m^{-1} \sum_{i=1}^m \mathbf{H}_i(\hat{\boldsymbol{\theta}})$ is the Hessian of the empirical likelihood function $\ell$ in (1) at $\hat{\boldsymbol{\theta}}$. Denote $\overline{\mathbf{H}}(\hat{\boldsymbol{\theta}}) := \widehat{\mathbf{H}}(\hat{\boldsymbol{\theta}}) - \mathbb{E}_{\boldsymbol{\theta}^*}[\widehat{\mathbf{H}}(\hat{\boldsymbol{\theta}})] = \widehat{\mathbf{H}}(\hat{\boldsymbol{\theta}}) - \mathbf{H}(\hat{\boldsymbol{\theta}})$.

First, we will use the uniform matrix Bernstein inequality (Lemma 4.5) to deduce that the empirical Hessian uniformly concentrates on its population expectation in the spectral norm. To do so, we need to verify the hypothesis of Lemma 4.5. First, by using Lemma 4.2, the entries in the Hessian $\mathbf{H}_i(\hat{\boldsymbol{\theta}})$ at any parameter $\hat{\boldsymbol{\theta}} \in \widehat{\boldsymbol{\Theta}}_0(\delta)$ are uniformly bounded by $J := C_{16}\left(\widetilde{C}_{16}/\delta\right)^{\mathrm{diam}(T)/2+4}$. By Gershgorin's circle theorem, it follows that the eigenvalues of the Hessian of $\mathbf{H}_i(\hat{\boldsymbol{\theta}})$ are uniformly bounded by $|E|J$ (as this is an upper bound of the row sums). Next, note that each entry of $\mathbf{H}_i(\hat{\boldsymbol{\theta}})^2$ is uniformly bounded by $|E|J^2$, so by a similar argument, the maximum eigenvalue of $\sum_{i=1}^m \mathbb{E}[\mathbf{H}_i(\hat{\boldsymbol{\theta}})^2]$ are uniformly bounded by $m|E|^2 J^2$. Lastly, for the Lipschitz continuity in (20), we may use the mean value theorem to deduce

$$
\|\mathbf{H}_i(\hat{\boldsymbol{\theta}}) - \mathbf{H}_i(\hat{\boldsymbol{\theta}}')\|_2 \leq |E|^2 \|\mathbf{H}_i(\hat{\boldsymbol{\theta}}) - \mathbf{H}_i(\hat{\boldsymbol{\theta}}')\|_{\max} \leq M|E|^2 \|\hat{\boldsymbol{\theta}} - \hat{\boldsymbol{\theta}}'\| \qquad \text{for all } \hat{\boldsymbol{\theta}}, \hat{\boldsymbol{\theta}}' \in \widehat{\boldsymbol{\Theta}}_0(\delta),
$$

where $M$ denotes the largest absolute value of the third-order derivatives of $\ell(\boldsymbol{\theta}; \sigma^{(i)})$ over all $\boldsymbol{\theta} \in \widehat{\boldsymbol{\Theta}}_0(\delta)$. According to Lemma 4.2(iii), $M \leq 4 \operatorname{diam}(T)(2c_4\delta)^{-4\operatorname{diam}(T)-2}$. Since the above bound holds almost surely for all parameters in the box, it also holds for the expected Hessians, and hence for their deviations from the mean.

Thus, by the uniform matrix Bernstein inequality (Lemma 4.5), we get

$$\mathbb{P}_{\boldsymbol{\theta}^*}\left(\sup_{\hat{\boldsymbol{\theta}} \in \widehat{\boldsymbol{\Theta}}_0(\delta)} \|\overline{\mathbf{H}}(\hat{\boldsymbol{\theta}})\|_2 \geq 1\right) \leq 2|E|\left(2\sqrt{|E|}(C_4 - c_4)\delta\right)^{|E|}\left(1 + \frac{16 \operatorname{diam}(T)}{(2c_4\delta)^{4\operatorname{diam}(T)+2}}\right)^{|E|} \exp\left(\frac{-m^2/8}{\sigma^2 + Rm/6}\right), \quad (33)$$

where $R = |E|J$ and $\sigma^2 = m|E|^2 J^2$. We will choose the sample size $m$ so that the right-hand side of (33) is at most $\varepsilon/2$. For this, it is sufficient to have

$$\log(2|E|) + |E| \log\left(2\sqrt{|E|}(C_4 - c_4)\delta(1 + 16 \operatorname{diam}(T)(2c_4\delta)^{-4\operatorname{diam}(T)-2})\right) + \frac{-m/8}{|E|^2 J^2 + |E|J/6} \leq \log \varepsilon/2.$$

Rearranging, it is enough to have

$$m \geq 8\left(1 + |E|C_{16}\left(\frac{\widetilde{C}_{16}}{\delta}\right)^{\frac{1}{2}\operatorname{diam}(T)+4}\right)^2 \quad (34)$$
$$\times \left[\log(2|E|) + |E| \log\left(2|E|^{1/2}(C_4 - c_4)\delta\left(1 + \frac{16 \operatorname{diam}(T)}{(2c_4\delta)^{4\operatorname{diam}(T)+2}}\right)\right) + \log(2/\varepsilon)\right].$$

We choose $C_{31}$ large enough so that $(C_{31}/\delta)^{\operatorname{diam}(T)+8}$ dominates the right-hand side above with the term $\log(2/\varepsilon)$ disregarded. Then we increase $C_{31}$ so the right-hand side above is dominated by $(C_{31}/\delta)^{\operatorname{diam}(T)+8} \log(1/\varepsilon)$, which is the sample complexity bound in (31).

Next, we will deduce that the extreme eigenvalues of $\widehat{\mathbf{H}}(\boldsymbol{\theta})$ are uniformly concentrated around the extreme eigenvalues of the expected Hessian over all parameters $\boldsymbol{\theta} \in \widehat{\boldsymbol{\Theta}}_0(\delta)$ with high probability. For the maximum eigenvalue, by Weyl's inequality for self-adjoint matrices,

$$\|\lambda_{\max}\left(\widehat{\mathbf{H}}(\hat{\boldsymbol{\theta}})\right) - \lambda_{\max}\left(\mathbb{E}[\widehat{\mathbf{H}}(\hat{\boldsymbol{\theta}})]\right)\| \leq \|\overline{\mathbf{H}}(\hat{\boldsymbol{\theta}})\|_2.$$

Thus (33) implies that $\lambda_{\max}\left(\widehat{\mathbf{H}}(\hat{\boldsymbol{\theta}})\right) \leq \lambda_{\max}\left(\mathbb{E}[\widehat{\mathbf{H}}(\hat{\boldsymbol{\theta}})]\right) + 1$ for all $\hat{\boldsymbol{\theta}} \in \widehat{\boldsymbol{\Theta}}_0(\delta)$ with probability at least $1 - (\varepsilon/2)$. By applying the same argument for $-\widehat{\mathbf{H}}(\hat{\boldsymbol{\theta}})$, we have that $\lambda_{\min}\left(\mathbf{H}(\hat{\boldsymbol{\theta}})\right) \geq \lambda_{\min}\left(\mathbb{E}[\mathbf{H}(\hat{\boldsymbol{\theta}})]\right) - 1$ for all $\hat{\boldsymbol{\theta}} \in \widehat{\boldsymbol{\Theta}}_0(\delta)$ with probability at least $1 - (\varepsilon/2)$. Hence by union bound and Lemma 4.4, we deduce the assertion (32). $\qquad\square$

Next, we prove Theorem 3.3, which we re-state below with more explicit constants.

**Theorem C.2** (Statistical estimation guarantee, Thm. 3.3). *Assume the hypothesis of Theorem 3.2 holds. Let $E_{32}$ denote the event in (32). Fix $\varepsilon \in (0, 1)$ and denote $\rho := \frac{C_{19}}{\delta} - 27$ and $C_7 := \frac{16\widetilde{C}_{19}}{C_{19}}$ Then we have*

$$\mathbb{P}_{\boldsymbol{\theta}^*}\left(E_{32} \cap \left\{\|\boldsymbol{\theta}^* - \hat{\boldsymbol{\theta}}^*\| \leq C_7\sqrt{|E|/m}\log(|E|/\varepsilon)\right\}\right) \geq 1 - 3\varepsilon$$

*provided that $m$ satisfies (31) and $m \geq \frac{|E|^2}{4C_{19}^6 c_4^6 \delta^3 \varepsilon}$.*

*Proof.* Fix $\varepsilon > 0$. By Theorem 3.2, $\mathbb{P}_{\boldsymbol{\theta}^*}(E_{32}) \geq 1 - \varepsilon$. At present let $R > 0$ be some yet-to-be-determined quantity.

We will show that the log-likelihood function $\ell$ attains a local maximizer within $Cm^{-1/2}$-ball around $\boldsymbol{\theta}^*$:

$$\mathbb{P}_{\boldsymbol{\theta}^*}\left(\sup_{\hat{\boldsymbol{\theta}} \in \widehat{\boldsymbol{\Theta}}_0(\delta),\, \|\hat{\boldsymbol{\theta}} - \boldsymbol{\theta}^*\| = Rm^{-1/2}} \ell(\hat{\boldsymbol{\theta}}; \sigma) - \ell(\boldsymbol{\theta}^*) < 0\right) \geq 1 - 2\varepsilon. \quad (35)$$

Hence by a union bound, the intersection of $E_{32}$ and the event in (35) occurs (under $\mathbb{P}_{\boldsymbol{\theta}^*}$) with probability at least $1 - 3\varepsilon$. On the event $E_{32}$, since $\ell$ is strictly concave over $\widehat{\boldsymbol{\Theta}}_0(\delta)$, it follows that the local maximizer near $\boldsymbol{\theta}^*$ must be

$\hat{\boldsymbol{\theta}}^*$, the global maximizer of $\ell$ over $\widehat{\boldsymbol{\Theta}}_0(\delta)$. This is enough to deduce (7). In the remainder of this proof we will write $\ell(\hat{\boldsymbol{\theta}}) = \frac{1}{m} \sum_{i=1}^m \ell(\hat{\boldsymbol{\theta}}; x_i)$ but write $\ell(\hat{\boldsymbol{\theta}}; x_i)$ to refer to a single sample.

Fix $\hat{\boldsymbol{\theta}} \in \widehat{\boldsymbol{\Theta}}_0(\delta)$ such that $\|\hat{\boldsymbol{\theta}} - \boldsymbol{\theta}^*\| = Rm^{-1/2}$. We introduce two random variables that we will bound to be small by using some concentration inequalities:

$$T_m(\hat{\boldsymbol{\theta}}) := \frac{\sqrt{m}}{\|\hat{\boldsymbol{\theta}} - \boldsymbol{\theta}^*\|} \left\langle \nabla_{\hat{\boldsymbol{\theta}}} \ell(\boldsymbol{\theta}^*) - \mathbb{E}\left[\nabla_{\hat{\boldsymbol{\theta}}} \ell(\boldsymbol{\theta}^*)\right], \hat{\boldsymbol{\theta}} - \boldsymbol{\theta}^* \right\rangle,$$

$$S_m(\hat{\boldsymbol{\theta}}) := \frac{1}{\|\hat{\boldsymbol{\theta}} - \boldsymbol{\theta}^*\|^2} \sup_{\mathbf{z} \in \widehat{\boldsymbol{\Theta}}_0(\delta)} \frac{1}{2} (\hat{\boldsymbol{\theta}} - \boldsymbol{\theta}^*)^T \widehat{\mathbf{H}}(\mathbf{z}) (\hat{\boldsymbol{\theta}} - \boldsymbol{\theta}^*),$$

where $\widehat{\mathbf{H}}$ denotes the Hessian of the $m$-sample log-likelihood function $\ell$ in (1). By first-order Taylor expansion,

$$\ell(\hat{\boldsymbol{\theta}}) - \ell(\boldsymbol{\theta}^*) \leq \langle \nabla_{\hat{\boldsymbol{\theta}}} \ell(\boldsymbol{\theta}^*), \hat{\boldsymbol{\theta}} - \boldsymbol{\theta}^* \rangle + \|\hat{\boldsymbol{\theta}} - \boldsymbol{\theta}^*\|^2 S_m(\hat{\boldsymbol{\theta}})$$

$$\leq \frac{\|\hat{\boldsymbol{\theta}} - \boldsymbol{\theta}^*\|}{\sqrt{m}} \cdot T_m(\hat{\boldsymbol{\theta}}) + \frac{1}{2} \|\hat{\boldsymbol{\theta}} - \boldsymbol{\theta}^*\|^2 \sup_{\mathbf{z} \in \widehat{\boldsymbol{\Theta}}_0(\delta)} \lambda_{\max}(\widehat{\mathbf{H}}(\mathbf{z})),$$

where the second inequality uses $\nabla_{\hat{\boldsymbol{\theta}}} \mathbb{E}[\ell(\boldsymbol{\theta}^*)] = 0$ and that $(\hat{\boldsymbol{\theta}} - \boldsymbol{\theta}^*)^T \widehat{\mathbf{H}}(\mathbf{z})(\hat{\boldsymbol{\theta}} - \boldsymbol{\theta}^*) \leq \lambda_{\max}(\widehat{\mathbf{H}}(\mathbf{z}))\|\hat{\boldsymbol{\theta}} - \boldsymbol{\theta}^*\|^2$. Dividing both sides by $\|\hat{\boldsymbol{\theta}} - \boldsymbol{\theta}^*\|^2 = R^2 m^{-1}$ and rearranging,

$$\frac{\ell(\hat{\boldsymbol{\theta}}) - \ell(\boldsymbol{\theta}^*)}{\|\hat{\boldsymbol{\theta}} - \boldsymbol{\theta}^*\|^2} \leq \frac{1}{R} \underbrace{\left(T_m(\hat{\boldsymbol{\theta}}) - \frac{R\rho}{4}\right)}_{=:I_1} + \underbrace{\left(\frac{1}{2} \sup_{\mathbf{z} \in \widehat{\boldsymbol{\Theta}}_0(\delta)} \lambda_{\max}(\widehat{\mathbf{H}}(\mathbf{z})) + \frac{\rho}{4}\right)}_{=:I_2} \tag{36}$$

where $\rho := \frac{C_{19}}{\delta} - 27$. We wish to show that the supremum of the left-hand side of (36) overall $\hat{\boldsymbol{\theta}} \in \widehat{\boldsymbol{\Theta}}_0(\delta)$ with $\|\hat{\boldsymbol{\theta}} - \boldsymbol{\theta}^*\| = Rm^{-1/2}$ is strictly negative with probability $1 - \varepsilon$. Indeed, by Theorem 3.2, we have

$$\mathbb{P}_{\boldsymbol{\theta}^*}(I_2 \leq -\rho/4) \geq 1 - 2\varepsilon$$

provided the sample complexity of (31). Thus it is enough to show that $I_1 < 0$ at least with probability $1 - \varepsilon$. Asymptotically as $m \to \infty$, $T_m(\hat{\boldsymbol{\theta}})$ is of order 1 with high probability by the central limit theorem. Below we give a non-asymptotic argument using Berry-Esseen theorem. We will show the following inequality

$$\mathbb{P}_{\boldsymbol{\theta}^*} \left( \sup_{\substack{\hat{\boldsymbol{\theta}} \in \widehat{\boldsymbol{\Theta}}_0(\delta) \\ \|\hat{\boldsymbol{\theta}} - \boldsymbol{\theta}^*\| = Rm^{-1/2}}} T_m(\hat{\boldsymbol{\theta}}) \geq \frac{R\rho}{4} \right) \leq |E| \exp\left(-\frac{\rho^2 R^2 \delta^2}{64|E|\widetilde{C}_{19}^2}\right) + \frac{3|E|}{8 C_{19}^3 c_4^3 \delta^{3/2} \sqrt{m}}. \tag{37}$$

Supposing (37) for the moment, Theorem 3.3 follows once we find $R$ (resp. $m$) large enough so that the first (resp. second) term on the right-hand side above are at most $\varepsilon$. The first term is smaller than $\varepsilon$ by simply rearranging (7) with $R = C_7 \sqrt{|E| \log(|E|/\varepsilon)}$. The second term is clearly satisfied by the choice of $m$ in the statement.

It remains to verify (37). To this end, write $\nabla_{\hat{\boldsymbol{\theta}}} \ell(\boldsymbol{\theta}^*) = \nabla_{\hat{\boldsymbol{\theta}}} \ell(\boldsymbol{\theta}^*) - \mathbb{E}\left[\nabla_{\hat{\boldsymbol{\theta}}} \ell(\boldsymbol{\theta}^*)\right] = \frac{1}{m} \sum_{i=1}^m \overline{U}_i$, where $\overline{U}_i := \nabla_{\boldsymbol{\theta}} \ell(\boldsymbol{\theta}^*; \sigma^{(i)}) \in \mathbb{R}^{|E|}$. We write $\overline{U}_i(k)$ for the $k^{\text{th}}$ coordinate of $\overline{U}_i$. Then by using Cauchy-Schwarz inequality and noting that $\|\boldsymbol{\theta} - \boldsymbol{\theta}^*\| = Cm^{-1/2}$, we get

$$T_m(\boldsymbol{\theta}) = \left\langle \frac{1}{\sqrt{m}} \sum_{i=1}^m \overline{U}_i, \frac{\hat{\boldsymbol{\theta}} - \boldsymbol{\theta}^*}{\|\hat{\boldsymbol{\theta}} - \boldsymbol{\theta}^*\|} \right\rangle \leq \left\| \frac{1}{\sqrt{m}} \sum_{i=1}^m \overline{U}_i \right\| = \sqrt{\sum_{k=1}^{|E|} \left| \frac{1}{\sqrt{m}} \sum_{i=1}^m \overline{U}_i(k) \right|^2}.$$

It is important to note that the distribution of the random variable on the last term above does not depend on $\hat{\boldsymbol{\theta}}$. Note that $\overline{U}_i$ for $i = 1, \ldots, n$ are independent mean zero i.i.d. random vectors in $\mathbb{R}^{|E|}$, as we will see below, their respective coordinates

have uniformly bounded variances. Hence by a union bond,

$$\mathbb{P}_{\boldsymbol{\theta}^*}\left(\sup_{\substack{\boldsymbol{\theta}\in\widehat{\Theta}_0(\delta)\\ \|\boldsymbol{\theta}-\boldsymbol{\theta}^*\|=m^{-1/2}}} T_m(\boldsymbol{\theta}) \geq t\right) \leq \sum_{k=1}^{|E|} \mathbb{P}_{\boldsymbol{\theta}^*}\left(\left|\frac{1}{\sqrt{m}}\sum_{i=1}^m \overline{U}_i(k)\right| \geq \frac{t}{\sqrt{|E|}}\right). \tag{38}$$

Let $Q_n^k = \frac{1}{\sqrt{m}}\sum_{i=1}^m \overline{U}_i(k)$. Then by the Berry-Esseen Theorem (Thm. 3.4.17 in (Durrett, 2019)) and the hypothesis, for $Z \sim N(0,1)$,

$$\sup_{z\in\mathbb{R}}\left|\mathbb{P}_{\boldsymbol{\theta}^*}\left(Q_n^k \leq \sigma z\right) - \mathbb{P}_{\boldsymbol{\theta}^*}\left(Z \leq z\right)\right| \leq \frac{3\mathbb{E}[|U_i(k)|^3]}{\sigma^3\sqrt{m}} \qquad \text{where} \qquad \sigma^2 = \sigma^2(k) = \mathrm{Var}_{\boldsymbol{\theta}^*}(\overline{U}_i(k)) = \mathbb{E}_{\boldsymbol{\theta}^*}[|\overline{U}_i(k)|^2].$$

Write $\sigma_{\min} = \min_k \sigma(k)$ and $\sigma_{\max} = \max_k \sigma(k)$. Combining with (38) and using a triangle inequality, we obtain

$$\mathbb{P}_{\boldsymbol{\theta}^*}\left(\sup_{\substack{\boldsymbol{\theta}\in\widehat{\Theta}_0(\delta)\\ \|\boldsymbol{\theta}-\boldsymbol{\theta}^*\|=m^{-1/2}}} T_m(\boldsymbol{\theta}) \geq t\right) \leq |E|\,\mathbb{P}\left(Z \geq \frac{t}{2\sqrt{|E|}\sigma_{\max}}\right) + \frac{1}{\sqrt{m}}\frac{3\sum_{k=1}^{|E|}\mathbb{E}_{\boldsymbol{\theta}^*}[|\overline{U}_1(k)|^3]}{\sigma_{\min}^3}. \tag{39}$$

To simplify the last term above, note that for each edge $e \in E$ with corresponding coordinate $k$ for $\overline{U}_i$, Lemma 4.2 and **A1** yield $|\overline{U}_i(k)| = \left|\frac{Z_x Z_y}{1+Z_x Z_y \hat{\theta}_e^*}\right| \leq 1/(2c_4\delta)$

$$\sum_{k=1}^{|E|}\mathbb{E}_{\boldsymbol{\theta}^*}[|U_i(k)|^3] \leq \frac{|E|}{(2c_4\delta)^3}.$$

Furthermore, using $\frac{\partial^2}{\partial\hat{\theta}_e^2}\ell(\boldsymbol{\theta}^*) = \frac{\partial}{\partial\hat{\theta}_e}\frac{Z_x Z_y}{1+Z_x Z_y \hat{\theta}_e} = -\left(\frac{Z_x Z_y}{1+Z_x Z_y \hat{\theta}_e}\right)^2 = -\left(\frac{\partial}{\partial\hat{\theta}_e}\ell(\hat{\boldsymbol{\theta}}^*)\right)^2 = -(\overline{U}_i(k))^2$ from Lemma 4.2**(i)** we see that

$$\sigma^2(k) = \mathrm{Var}_{\boldsymbol{\theta}^*}(\overline{U}_i(k)) = \mathbb{E}_{\boldsymbol{\theta}^*}\left[U_i(k)^2\right] = -\mathbb{E}_{\boldsymbol{\theta}^*}\left[\frac{\partial^2}{\partial\hat{\theta}_e^2}\ell(\boldsymbol{\theta}^*)\right] \in \left[\frac{C_{19}}{\delta}, \frac{\widetilde{C}_{19}}{\delta}\right].$$

To get the last inclusion above we used Lemma 4.4. Going back to (39), this shows

$$\mathbb{P}_{\boldsymbol{\theta}^*}\left(\sup_{\substack{\boldsymbol{\theta}\in\widehat{\Theta}_0(\delta)\\ \|\boldsymbol{\theta}-\boldsymbol{\theta}^*\|=m^{-1/2}}} T_m(\boldsymbol{\theta}) \geq t\right) \leq |E|\,\mathbb{P}\left(Z \geq \frac{t\delta}{2\sqrt{|E|}\tilde{C}_{19}}\right) + \frac{1}{\sqrt{m}}\frac{3|E|}{8C_{19}^3 c_4^3 \delta^{3/2}}$$

$$\leq |E|\exp\left(-\frac{t^2\delta^2}{4|E|\widetilde{C}_{19}^2}\right) + \frac{1}{\sqrt{m}}\frac{3|E|}{8C_{19}^3 c_4^3 \delta^{3/2}}$$

where we used the standard Gaussian tail bound $\mathbb{P}(Z \geq x) \leq \exp(-x^2/2)$. Setting $t = \frac{R\rho}{4}$ gives (37), as desired. $\qquad\square$

In the remainder of this section, we prove Theorem 3.4.

**Theorem C.3** (Statistical and computational estimation guarantee, Thm. 3.4). *Suppose the hypothesis of Theorem 3.3 holds. Let $(\hat{\boldsymbol{\theta}}_k)_{k\geq 0}$ denote the sequence of estimated parameters generated by the coordinate maximization algorithm (see Alg. 1) with the initial estimate $\hat{\boldsymbol{\theta}}_0$ satisfying*

$$\|\hat{\boldsymbol{\theta}}_0 - \boldsymbol{\theta}^*\| \leq \frac{(C_{19} - 27\delta)C_{40}}{C_{19} + \widetilde{C}_{19}}\frac{\delta}{2}, \tag{40}$$

*where $C_{40} := (C_4 - C_3) \wedge (c_4 - c_3) > 0$. Then with probability at least $1 - 3\varepsilon$, for all $k \geq 0$,*

$$\|\hat{\boldsymbol{\theta}}^* - \hat{\boldsymbol{\theta}}_k\|^2 \leq \frac{\widetilde{C}_{19} - 27\delta}{C_{19} - 27\delta} \left(1 - \frac{C_{19}\delta^{-1} - 27}{\widetilde{C}_{19}\delta^{-1} - 27}\right)^{k-1} \|\hat{\boldsymbol{\theta}}^* - \hat{\boldsymbol{\theta}}_0\|^2. \tag{41}$$

*In particular,*

$$\|\boldsymbol{\theta}^* - \hat{\boldsymbol{\theta}}_k\| \leq \underbrace{C_7\sqrt{|E|/m}\log(|E|/\varepsilon)}_{\text{=statistical error}} + \underbrace{\sqrt{\frac{\widetilde{C}_{19} - 27\delta}{C_{19} - 27\delta}} \left(1 - \frac{C_{19}\delta^{-1} - 27}{\widetilde{C}_{19}\delta^{-1} - 27}\right)^{(k-1)/2} \|\hat{\boldsymbol{\theta}}^* - \hat{\boldsymbol{\theta}}_0\|}_{\text{=computational error}}. \tag{42}$$

A crucial ingredient is the following local confinement result of coordinate maximization for the maximum likelihood landscape.

**Lemma C.4** (Local confinement of coordinate maximization)**.** *Suppose the hypothesis of Theorem 3.3 holds. Fix $\varepsilon \in (0, 1)$ and let $E_7$ denote the event in (7). If $(\hat{\boldsymbol{\theta}}_k)_{k \geq 0}$ is a sequence of parameters computed by the coordinate maximization algorithm (Alg. 1) with the initial estimate $\hat{\boldsymbol{\theta}}_0$ satisfying (40), then*

$$\mathbb{P}_{\boldsymbol{\theta}^*}\left(E_7 \cap \left\{\hat{\boldsymbol{\theta}}_k \in \text{Int}\left(\widehat{\boldsymbol{\Theta}}_0(\delta)\right) \text{ for all } k \geq 0\right\}\right) \geq 1 - 3\varepsilon, \tag{43}$$

*provided $m \geq 1$ is large enough so that*

$$\frac{C_7\sqrt{|E|\log|E|}}{\sqrt{m}} < \frac{C_{40}}{2}\frac{C_{19}}{\widetilde{C}_{19} + C_{19}}\delta. \tag{44}$$

*Proof.* In this proof, we denote $f = -\ell$ for the negative of the empirical log-likelihood in (1). We will crucially use the fact that coordinate minimization generates a sequence of iterates that monotonically decreases the objective $f$. Namely, let $A_0$ denote the event of interest in (43). Define the following events:

$$A_1 := \left\{\{\hat{\boldsymbol{\theta}} \in \widehat{\boldsymbol{\Theta}}_0(\delta) \mid f(\hat{\boldsymbol{\theta}}) \leq f(\hat{\boldsymbol{\theta}}_0)\} \subseteq \text{Int}\left(\widehat{\boldsymbol{\Theta}}_0(\delta)\right)\right\},$$

$$A_2 := \left\{\|\hat{\boldsymbol{\theta}}^* - \boldsymbol{\theta}^*\| < \frac{C_{40}}{2}\frac{C_{19}}{\widetilde{C}_{19} + C_{19}}\delta\right\},$$

$$E_{32} := \text{the event in (32) in Theorem 3.2.}$$

Note that both $f$ and $\hat{\boldsymbol{\theta}}^*$ are random and so both $A_1$ and $A_2$ are indeed events. We claim that the following inclusions hold:

$$E_7 \quad \subseteq \quad A_2 \cap E_{32} \quad \subseteq \quad A_1 \cap E_{32} \quad \subseteq \quad A_0.$$

The first inclusion is immediate by the definition of the events and the choice of sample size (44). Since $\mathbb{P}_{\boldsymbol{\theta}^*}(E_7) \geq 1 - 3\varepsilon$ by Theorem 3.3, this implies that $A_0$ occurs with probability at least $1 - 3\varepsilon$, as desired. We first show $A_1 \subseteq A_0$ by induction. For the base step, note that, by **A1**, $\boldsymbol{\theta}^* \in \boldsymbol{\Theta}_0(\delta)$ and that $\boldsymbol{\Theta}_0(\delta)$ is a closed box contained in the $C_{40}\delta$-interior of the closed box $\widehat{\boldsymbol{\Theta}}_0(\delta)$ (see **A1**). Hence,

$$\left\{\tilde{\boldsymbol{\theta}} \,:\, \|\tilde{\boldsymbol{\theta}} - \boldsymbol{\theta}^*\| < C_{40}\delta\right\} \subseteq \text{Int}\left(\widehat{\boldsymbol{\Theta}}_0(\delta)\right). \tag{45}$$

Since the initial estimate $\hat{\boldsymbol{\theta}}_0$ satisfies (40), in particular it holds that $\|\hat{\boldsymbol{\theta}}_0 - \boldsymbol{\theta}^*\| < C_{40}\delta$. Hence from the above inclusion, we have $\hat{\boldsymbol{\theta}}_0 \in \text{Int}\left(\widehat{\boldsymbol{\Theta}}_0(\delta)\right)$. This verifies the base step.

For the induction step, suppose $A_1 \cap E_{32}$ holds and $\hat{\boldsymbol{\theta}}_0, \ldots, \hat{\boldsymbol{\theta}}_k \in \text{Int}\left(\widehat{\boldsymbol{\Theta}}_0(\delta)\right)$ for some $k \geq 0$. We wish to show that $\hat{\boldsymbol{\theta}}_{k+1} \in \text{Int}\left(\widehat{\boldsymbol{\Theta}}_0(\delta)\right)$. Recall that $\hat{\boldsymbol{\theta}}_{k+1}$ is obtained from $\hat{\boldsymbol{\theta}}_k$ by updating a coordinate $\hat{\theta}_{k;e}$ corresponding to some unique edge $e$ in $T$. We now use the fact that the restriction of $f$ on each coordinate is strictly convex almost surely (Fukami & Tateno, 1989). Denote this function $\theta_e \mapsto f(\boldsymbol{\theta})$ by $f_e$. Then by definition

$$\hat{\theta}_{k+1;e} = \arg\min_{\theta \in [-1,1]} f_e(\theta).$$

In addition, define

$$\tilde{\theta}_{k+1;e} = \underset{\theta \in [1 - 2C_4 \delta, 1 - 2c_4 \delta]}{\arg\min} f_e(\theta)$$

and let $\tilde{\boldsymbol{\theta}}_{k+1}$ denote the parameter obtained from $\hat{\boldsymbol{\theta}}_k$ by replacing $\hat{\theta}_{k;e}$ with $\tilde{\theta}_{k+1;e}$. By induction hypothesis, $\hat{\boldsymbol{\theta}}_k \in \text{Int}\left(\widehat{\boldsymbol{\Theta}}_0(\delta)\right)$. In particular, $\hat{\theta}_{k;e} \in (1 - 2C_4\delta, 1 - 2c_4\delta)$. Hence $f(\tilde{\boldsymbol{\theta}}_{k+1}) \le f(\hat{\boldsymbol{\theta}}_k) \le \cdots \le f(\hat{\boldsymbol{\theta}}_0)$. Thus, on the event $A_1$, we deduce $\tilde{\boldsymbol{\theta}}_{k+1} \in \text{Int}\left(\widehat{\boldsymbol{\Theta}}_0(\delta)\right)$. This implies $\tilde{\theta}_{k+1;e}$ is in fact in the open interval $(1 - 2C_4\delta, 1 - 2c_4\delta)$. Now since $f_e$ is strictly convex on the whole interval $[-1, 1]$, it follows that $\tilde{\theta}_{k+1;e} = \hat{\theta}_{k+1;e}$; in words, the strictly convex function $f_e$ attains its global minimizer in the whole domain $[-1, 1]$ inside the open interval $(1 - 2C_4\delta, 1 - 2c_4\delta)$. This yields $\hat{\boldsymbol{\theta}}_{k+1} = \tilde{\boldsymbol{\theta}}_{k+1} \in \text{Int}\left(\widehat{\boldsymbol{\Theta}}_0(\delta)\right)$, which completes the induction step. This shows the inclusion $A_1 \cap E_{32} \subseteq A_0$.

It remains to show $A_2 \cap E_{32} \subseteq A_1$ as it is obviously a subset of $E_{32}$. To this effect, we suppose that $A_2 \cap E_{32}$ holds and choose $\boldsymbol{\theta} \in \widehat{\boldsymbol{\Theta}}_0(\delta)$ with $f(\boldsymbol{\theta}) \le f(\hat{\boldsymbol{\theta}}_0)$. We wish to show that $\boldsymbol{\theta} \in \text{Int}\left(\widehat{\boldsymbol{\Theta}}_0(\delta)\right)$. By (45), it suffices to show that $\|\boldsymbol{\theta} - \boldsymbol{\theta}^*\| < C_{40}\delta$. Note that on the event $A_2$, we have that $\hat{\boldsymbol{\theta}}^*$ is in the interior of $\widehat{\boldsymbol{\Theta}}_0(\delta)$ by (45). Since $\hat{\boldsymbol{\theta}}^*$ is the global maximizer of the empirical log-likelihood $\ell$ over the box $\widehat{\boldsymbol{\Theta}}_0(\delta)$, it follows that $\nabla f(\hat{\boldsymbol{\theta}}^*) = 0$ on $A_2$. Let $\mu := \frac{C_{19}}{\delta} - 27$ and $L := \frac{\widetilde{C}_{19}}{\delta} + 27$. On the event $E_{32}$, $f$ is $\mu$-strongly convex and $L$-smooth over $\widehat{\boldsymbol{\Theta}}_0(\delta)$. On this event, $f$ restricted on $\widehat{\boldsymbol{\Theta}}_0(\delta)$ is upper and lower bounded by quadratic functions as

$$f(\hat{\boldsymbol{\theta}}^*) + \frac{\mu}{2}\|\boldsymbol{\theta} - \hat{\boldsymbol{\theta}}^*\|^2 \le f(\boldsymbol{\theta}) \le f(\hat{\boldsymbol{\theta}}^*) + \frac{L}{2}\|\boldsymbol{\theta} - \hat{\boldsymbol{\theta}}^*\|^2 \quad \text{for all } \boldsymbol{\theta} \in \widehat{\boldsymbol{\Theta}}_0(\delta).$$

Hence it follows that, for each $\boldsymbol{\theta} \in \widehat{\boldsymbol{\Theta}}_0(\delta)$, $f(\boldsymbol{\theta}) \le f(\hat{\boldsymbol{\theta}}_0)$ implies $\frac{\mu}{2}\|\boldsymbol{\theta} - \hat{\boldsymbol{\theta}}^*\|^2 \le \frac{L}{2}\|\hat{\boldsymbol{\theta}}_0 - \hat{\boldsymbol{\theta}}^*\|^2$. Using the hypothesis (40), this yields

$$
\begin{aligned}
\|\boldsymbol{\theta} - \boldsymbol{\theta}^*\| &\le \|\hat{\boldsymbol{\theta}}^* - \boldsymbol{\theta}^*\| + \|\boldsymbol{\theta} - \hat{\boldsymbol{\theta}}^*\| \\
&\le \|\hat{\boldsymbol{\theta}}^* - \boldsymbol{\theta}^*\| + \frac{L}{\mu}\|\hat{\boldsymbol{\theta}}_0 - \hat{\boldsymbol{\theta}}^*\| \\
&\le \frac{L + \mu}{\mu}\left(\|\hat{\boldsymbol{\theta}}^* - \boldsymbol{\theta}^*\| + \|\hat{\boldsymbol{\theta}}_0 - \boldsymbol{\theta}^*\|\right) \\
&\le \frac{\widetilde{C}_{19} + C_{19}}{C_{19} - 27\delta}\left(\frac{C_{40}}{2}\frac{C_{19} - 27\delta}{\widetilde{C}_{19} + C_{19}}\delta + \|\hat{\boldsymbol{\theta}}_0 - \boldsymbol{\theta}^*\|\right) \\
&\le C_{40}\delta.
\end{aligned}
$$

According to (45), this implies $\boldsymbol{\theta} \in \text{Int}\left(\widehat{\boldsymbol{\Theta}}_0(\delta)\right)$ as desired. $\qquad \square$

We now prove Theorem 3.4, which we re-state below with a more explicit dependence on constants. In the proof, we will use a general result (Lemma 4.6) on linear convergence of coordinate minimization algorithm for minimizing strongly convex and smooth objectives with a closed constraint set. For the flow of the paper, we relegated its statement and proof in Section D.

***Proof of Theorem 3.4/C.3.*** Let $E_{43}$ denote the event in (43). By Lemma C.4, $\mathbb{P}_{\boldsymbol{\theta}^*}(E_{43}) \ge 1 - 3\varepsilon$. On the event $E_{43}$, the iterates $(\hat{\boldsymbol{\theta}}_k)_{k \ge 0}$ are confined in the interior of the box constraint set $\widehat{\boldsymbol{\Theta}}_0(\delta)$. Since the log-likelihood function $\ell$ is strictly convex in each coordinate, it follows that if we had imposed the additional open box constraint $\text{Int}(\widehat{\boldsymbol{\Theta}}_0(\delta))$ when we computed the coordinate maximization iterates in Alg. 1, the same iterates would have been generated. Thus, without loss of generality, we will assume that the iterates $(\hat{\boldsymbol{\theta}}_k)_{k \ge 0}$ are computed via the coordinate maximization algorithm for the following constrained maximization problem:

$$\max_{\boldsymbol{\theta} \in \text{Int}(\widehat{\boldsymbol{\Theta}}_0(\delta))} \ell(\boldsymbol{\theta}). \tag{46}$$

On the event $E_{43}$, $\ell$ is $\rho := (C_{19}\delta^{-1} - 27)$ - strongly concave and $L := (\widetilde{C}_{19}\delta^{-1} + 27)$-smooth over $\widehat{\Theta}_0(\delta)$. Hence we can apply Lemma 4.6 with $\rho = (C_{19}\delta^{-1} - 27)$ and $(L_i; i \in |E|)$ are any uniform (over $\widehat{\Theta}_0(\delta)$) upper-bounds for corresponding diagonal entry of the empirical Hessian on $E_{43}$. Clearly,

$$\max_i L_i \leq \sup_{\hat{\boldsymbol{\theta}} \in \widehat{\Theta}_0(\delta)} \sup_{\boldsymbol{v} \in S^{|E|-1}} \langle \boldsymbol{v}, \widehat{\mathbf{H}}(\hat{\boldsymbol{\theta}})\boldsymbol{v} \rangle \leq L.$$

Therefore, we may choose $L_{\max} = L$. This gives, for all $k \geq 0$,

$$\frac{\rho}{2}\|\boldsymbol{\theta} - \hat{\boldsymbol{\theta}}^*\|^2 \leq \ell(\boldsymbol{\theta}^*) - \ell(\boldsymbol{\theta}_k) \leq \left(1 - \frac{\rho}{L}\right)^{k-1}(\ell(\hat{\boldsymbol{\theta}}^*) - \ell(\hat{\boldsymbol{\theta}}_0)) \leq \frac{L}{2}\left(1 - \frac{\rho}{L}\right)^{k-1}\|\hat{\boldsymbol{\theta}}^* - \boldsymbol{\theta}\|^2.$$

From this, we can deduce (41) immediately. Lastly, (42) follows by combining (41) and Theorem 3.3 with triangle inequality. $\qquad\square$

## D. Coordinate minimization for constrained strongly convex and smooth problems

It is a classcial result in optimization by Beck and Tehruashvili (Beck & Tetruashvili, 2013) that alternating minimization (two-block unconstrained coordinate minimization) converges to the global minimizer at a linear rate $1 - \frac{\rho}{\min\{L_1, L_2\}}$ for $\rho$-strongly convex and blockwise $(L_1, L_2)$-smooth objectives. It is straightforward to extend this result for multi-block unconstrained coordinate minimization to obtain linear convergence with rate $1 - \frac{\rho}{\min\{L_1, \ldots, L_b\}}$, where $f$ is $b$-block smooth with block-smoothness parameters $L_1, \ldots, L_b$. This result is stated and proved below.

*Proof of Lemma 4.6.* We follow the approach of (Beck & Tetruashvili, 2013) and write

$$\boldsymbol{\theta}_t = (\theta_{n;1}, \cdots, \theta_{n;i}, \theta_{n-1,i+1}, \cdots, \theta_{n-1,b}) \qquad \text{for } t = n + \frac{i}{b} \qquad \text{for } i \in \{0, 1, , b-1\}.$$

We will always write $t = n + i/b$ for some $n \geq 0$ and some $i \in \{0, 1, , b-1\}$ in the sequel.

It follows from Lemma 5.1 in (Beck & Tetruashvili, 2013) that

$$f(\boldsymbol{\theta}_t) - f(\boldsymbol{\theta}_{t+1/b}) \geq \frac{1}{2L_{i+1}}\|\nabla f(\boldsymbol{\theta}_t)\|^2 \qquad \text{whenever} \qquad t = n + \frac{i}{b}.$$

Hence, whenever $t = n + i/b$ for some $i \in \{0, 1, , b-1\}$:

$$f(\boldsymbol{\theta}_t) - f(\boldsymbol{\theta}_{t+1}) \geq f(\boldsymbol{\theta}_t) - f(\boldsymbol{\theta}_{t+1/b}) \geq \frac{1}{2L_{i+1}}\|\nabla f(\boldsymbol{\theta}_t)\|^2. \tag{47}$$

Since $f$ is convex, with unique minimum $\boldsymbol{\theta}^*$, we can apply the Cauchy-Schwarz inequality to get

$$f(\boldsymbol{\theta}_t) - f^* \leq \langle \nabla f(\boldsymbol{\theta}_t), \boldsymbol{\theta}_t - \boldsymbol{\theta}^* \rangle \leq \|\nabla f(\boldsymbol{\theta}_t)\|\|\boldsymbol{\theta}_k - \boldsymbol{\theta}^*\| \qquad \text{for all } t.$$

Moreover, since $\boldsymbol{\theta}_t$ is attained by repeated minimization problems $f(\boldsymbol{\theta}_t) \leq f(\boldsymbol{\theta}_0)$ and so

$$f(\boldsymbol{\theta}_k) - f^* \leq C\|\nabla f(\boldsymbol{\theta}_k)\| \qquad \text{where } C = \sup\{\|\boldsymbol{\theta} - \boldsymbol{\theta}^*\| : f(\boldsymbol{\theta}) \leq f(\boldsymbol{\theta}_0)\} < \infty.$$

Combining this with (47) we have

$$f(\boldsymbol{\theta}_t) - f(\boldsymbol{\theta}_{t+1}) \geq \frac{(f(\boldsymbol{\theta}_t) - f^*)^2}{2L_{i+1}C^2} \qquad \text{whenever } t = n + \frac{i}{b}.$$

If we set $a_n = a_n(i) = f(\boldsymbol{\theta}_{n+i/b}) - f^*$ then

$$a_n - a_{n+1} = f(\boldsymbol{\theta}_{n+i/b}) - f(\boldsymbol{\theta}_{n+1+i/b}) \geq \frac{(f(\boldsymbol{\theta}_{n+i/b}) - f^*)^2}{2L_{i+1}C} = \gamma a_n^2 \qquad \text{where } \gamma := \frac{1}{2L_{i+1}C^2}.$$

By Lemma 3.5 in (Beck & Tetruashvili, 2013), we conclude $a_n \leq \frac{1}{\gamma n} = \frac{2L_{i+1}C^2}{n}$ for all $n \geq 1$. Next, by strong convexity, and for $t = n + i/b$

$$a_n(i) = f(\boldsymbol{\theta}_t) - f^* \leq \frac{1}{2\rho}\|\nabla f(\boldsymbol{\theta}_t)\|^2 \qquad\qquad \text{(strong convexity)}$$

$$\leq \frac{L_{i+1}}{\sigma}(f(\boldsymbol{\theta}_t) - f(\boldsymbol{\theta}_{t+1})) = \frac{L_{i+1}}{\sigma}(a_n(i) - a_{n+1}(i)) \qquad \text{(by (47))}.$$

Upon rearranging,

$$a_{n+1}(i) = f(\boldsymbol{\theta}_{n+1+i/b}) - f^* \leq \left(1 - \frac{\rho}{L_{i+1}}\right)\left(f(\boldsymbol{\theta}_{n+i/b}) - f^*\right) = \left(1 - \frac{\rho}{L_{i+1}}\right)a_n(i)$$

and so, by induction,

$$f(\boldsymbol{\theta}_{n+i/b}) - f^* \leq \left(1 - \frac{\rho}{L_{i+1}}\right)^n (f(\boldsymbol{\theta}_{i/b}) - f^*) \leq \left(1 - \frac{\rho}{L_{i+1}}\right)^n (f(\boldsymbol{\theta}_0) - f^*).$$

Now just choose the index $i^*$ such that $L^* = L_{i^*} = \min L_i$ so that

$$f(\boldsymbol{\theta}_{n+1}) - f^* \leq f(\boldsymbol{\theta}_{n+(i^*-1)/b}) - f^* \leq \left(1 - \frac{\rho}{L^*}\right)^n (f(\boldsymbol{\theta}_0) - f^*).$$

This is simply re-indexing the desired statement. $\qquad\square$

## E. Almost Sure Statements for the Hessian

In this section we prove Lemma 4.2, noting that the first two are contained in (Clancy, Jr. et al., 2025a) and so we just need to show (15) holds. Maintaining the notation from the Hessian, we see that (Clancy, Jr. et al., 2025a) essentially computed

$$\frac{\partial}{\partial\hat{\theta}_f}Z_y = Z_v \prod_{j=1}^N \hat{\theta}_{\{y_j,y_{j-1}\}} \prod_{j=0}^N \frac{1 - (\hat{\theta}_{\{y_j,w_j\}}Z_{w_j})^2}{\left(1 + \hat{\theta}_{\{y_j,w_j\}}\hat{\theta}_{\{y_j,y_{j-1}\}}Z_{w_j}Z_{y_{j-1}}\right)^2}$$

provided that the edge $f$ is contained in the subtree $T_y$. In particular, since each of the denominators is at least $(2c_4\delta)^2$, we get the following claim:

**Claim 1.** *Let $T_y$ be a binary tree rooted at $y$ and let $d$ denote the maximum distance from $y$ to a leaf in $T_y$. Then*

$$\left|\frac{\partial}{\partial\hat{\theta}_f}Z_y\right| \leq \frac{1}{(2c_4\delta)^{2d}}.$$

*Proof of Lemma 4.2.* This bound trivially holds for all $e_1 = e_2 = e_3$, and using Lemma 4.2**(i,ii)** it is also easily checked (using the symmetry of mixed partial derivatives) if $e_i = e_j = e$ and $e_k = f$ for distinct $i, j, k$ since

$$\frac{\partial^3}{\partial\hat{\theta}_f\partial\hat{\theta}_e^2}\ell(\hat{\boldsymbol{\theta}};\sigma) = \frac{\partial^2}{\partial\hat{\theta}_f\partial\hat{\theta}_e}\frac{Z_xZ_y}{(1+\hat{\theta}_eZ_xZ_y)^2} = \frac{\partial}{\partial\hat{\theta}_f}\frac{-2Z_x^2Z_y^2}{(1+\hat{\theta}_eZ_xZ_y)^3}$$

$$= \frac{-2Z_x^2Z_y(2-\hat{\theta}_eZ_xZ_y)}{(1+\hat{\theta}_eZ_xZ_y)^4}\frac{\partial}{\partial\hat{\theta}_f}Z_y.$$

So, using Claim 1 and $d$ the maximum distance from $y$ to a leaf in $T_y$,

$$\left|\frac{\partial^3}{\partial\hat{\theta}_f\partial\hat{\theta}_e^2}\ell(\hat{\boldsymbol{\theta}};\sigma)\right| \leq \frac{6}{(2c_4\delta)^4}\frac{1}{(2c_4\delta)^{2d}} = 6(2c_4\delta)^{-2\,\mathrm{diam}(T)-4}.$$

Therefore, we just check when $e_1, e_2, e_3$ are distinct. There are two cases to consider (again using symmetry of mixed partials). Say $e_3 = \{x, y\}$. Either $e_1 \in T_x$ and $e_2 \in T_y$ or both $e_1, e_2 \in T_y$.

In the case of the former, we see

$$\frac{\partial^3}{\partial\hat{\theta}_{e_1}\partial\hat{\theta}_{e_2}\partial\hat{\theta}_{e_3}}\ell(\hat{\boldsymbol{\theta}};\sigma) = \frac{\partial^2}{\partial\hat{\theta}_{e_1}\partial\hat{\theta}_{e_2}}\frac{Z_xZ_y}{1+\hat{\theta}_{e_3}Z_xZ_y} = \frac{1}{(1+\hat{\theta}_{e_3}Z_xZ_y)^2}\left(Z_x\frac{\partial}{\partial\hat{\theta}_{e_1}}Z_x + Z_y\frac{\partial}{\partial\hat{\theta}_{e_2}}Z_y\right)$$

and, since the sum of distance from $x$ to any leaf in $T_x$ and the distance from $y$ to any leaf in $T_y$ is at most the diameter,

$$\left|\frac{\partial^3}{\partial\hat{\theta}_{e_1}\partial\hat{\theta}_{e_2}\partial\hat{\theta}_{e_3}}\ell(\hat{\boldsymbol{\theta}};\sigma)\right| = \frac{1}{(2c_4\delta)^2}\frac{2}{(2c_4\delta)^{\operatorname{diam}(T)}}.$$

The latter case is a bit harder. In this case we will assume that $e_3 = e = \{x,y\}$ and $e_2 = f = \{u,v\}$ as in Lemma 4.2. Thus

$$\frac{\partial^3}{\partial\hat{\theta}_{e_1}\partial\hat{\theta}_{e_2}\partial\hat{\theta}_{e_3}}\ell(\hat{\boldsymbol{\theta}};\sigma) = \frac{\partial}{\partial\hat{\theta}_{e_1}}\frac{Z_xZ_v}{(1+\hat{\theta}_eZ_xZ_y)^2}\prod_{j=1}^N\hat{\theta}_{\{y_j,y_{j-1}\}}\prod_{j=0}^N\frac{\left(1-(\hat{\theta}_{\{y_j,w_j\}}Z_{w_j})^2\right)}{\left(1+\hat{\theta}_{\{y_j,w_j\}}\hat{\theta}_{\{y_j,y_{j-1}\}}Z_{w_j}Z_{y_{j-1}}\right)^2}$$

Note, we can suppose that $e_1 \neq \{y_j,y_{j-1}\}$ for any of the edges on the path from $e$ to $f$ as well as also that $e_1 \notin T_v$ as these are covered by the previous case. Therefore either $e_1 = \{y_k,w_k\}$ for some $k$ or $e_1 \in T_{w_k}$ for some $k$. If it is an edge $e_1 = \{y_k,w_k\}$ then $Z_{y_j}$ depends on $\hat{\theta}_{e_1}$ for $j \geq k$ but none of the other magnetizations appearing in the right-hand side of the above equation; while if $e_1 \in T_{w_k}$ then there is the additional dependence on $Z_{w_k}$. Note that for any $N \geq j > k$ and $e_1 = \{y_k,w_k\}$ it holds that

$$\frac{\partial}{\partial\hat{\theta}_{e_1}}\frac{1-(\hat{\theta}_{\{y_j,w_j\}}Z_{w_j})^2}{(1+\hat{\theta}_{\{y_j,w_j\}}\hat{\theta}_{\{y_j,y_{j-1}\}}Z_{w_j}Z_{y_{j-1}})^2}$$

$$= \frac{2\left(1-(\hat{\theta}_{\{y_j,w_j\}}Z_{w_j})^2\right)\hat{\theta}_{\{y_j,w_j\}}\hat{\theta}_{\{y_j,y_{j-1}\}}Z_{w_j}\left(1-(\hat{\theta}_{\{y_j,w_j\}}\hat{\theta}_{\{y_j,y_{j-1}\}}Z_{w_j})^2\right)}{(1+\hat{\theta}_{\{y_j,w_j\}}\hat{\theta}_{\{y_j,y_{j-1}\}}Z_{w_j}Z_{y_{j-1}})^3}\frac{\partial}{\partial\hat{\theta}_{e_1}}Z_{y_j}$$

$$= \frac{1-(\hat{\theta}_{\{y_j,w_j\}}Z_{w_j})^2}{(1+\hat{\theta}_{\{y_j,w_j\}}\hat{\theta}_{\{y_j,y_{j-1}\}}Z_{w_j}Z_{y_{j-1}})^2}E_1 \qquad \text{(say)}$$

as well as

$$\frac{\partial}{\partial\hat{\theta}_{e_1}}\frac{1-(\hat{\theta}_{\{y_k,w_k\}}Z_{w_k})^2}{(1+\hat{\theta}_{\{y_k,w_k\}}\hat{\theta}_{\{y_k,y_{k-1}\}}Z_{w_k}Z_{y_{k-1}})^2} = -\frac{2Z_{w_k}(\hat{\theta}_{\{y_k,w_k\}}Z_{w_k}+\hat{\theta}_{\{y_k,y_{k-1}\}}Z_{y_{k-1}})}{(1+\hat{\theta}_{\{y_k,w_k\}}\hat{\theta}_{\{y_k,y_{k-1}\}}Z_{w_k}Z_{y_{k-1}})^3}$$

$$= \frac{1-(\hat{\theta}_{\{y_k,w_k\}}Z_{w_k})^2}{(1+\hat{\theta}_{\{y_k,w_k\}}\hat{\theta}_{\{y_k,y_{k-1}\}}Z_{w_k}Z_{y_{k-1}})^2}E_2$$

$$\frac{\partial}{\partial\hat{\theta}_{e_1}}\frac{Z_xZ_v}{(1+\hat{\theta}_eZ_xZ_y)^2} = \frac{-2Z_x^2Z_vZ_y}{(1+\hat{\theta}_eZ_xZ_y)^3}\frac{\partial}{\partial e_1}Z_y = \frac{Z_xZ_v}{(1+\hat{\theta}_eZ_xZ_y)^2}E_3 \text{ (say)}.$$

Here $E_1, E_2, E_3$ are (signed) errors satisfying

$$|E_1| \leq \frac{2}{(2c_4\delta)} \times \frac{1}{(2c_4\delta)^{2\operatorname{diam}(T)}}$$

$$|E_2| \leq \frac{4}{(2c_4\delta)^2}$$

$$|E_3| \leq \frac{2}{(2c_4\delta)} \times \frac{1}{(2c_4\delta)^{2\operatorname{diam}(T)}}.$$

Therefore by the product rule when $e_1 = \{y_k, w_k\}$ then

$$
\left| \frac{\partial^3}{\partial \hat{\theta}_{e_1} \partial \hat{\theta}_{e_2} \partial \hat{\theta}_{e_3}} \ell(\hat{\boldsymbol{\theta}}; \sigma) \right| = \left| \frac{\partial}{\partial \hat{\theta}_{e_1}} \frac{Z_x Z_v}{(1 + \hat{\theta}_e Z_x Z_y)^2} \prod_{j=1}^{N} \hat{\theta}_{\{y_j, y_{j-1}\}} \prod_{j=0}^{N} \frac{\left(1 - (\hat{\theta}_{\{y_j, w_j\}} Z_{w_j})^2\right)}{\left(1 + \hat{\theta}_{\{y_j, w_j\}} \hat{\theta}_{\{y_j, y_{j-1}\}} Z_{w_j} Z_{y_{j-1}}\right)^2} \right|
$$

$$
= \left| \frac{Z_x Z_v}{(1 + \hat{\theta}_e Z_x Z_y)^2} \prod_{j=1}^{N} \hat{\theta}_{\{y_j, y_{j-1}\}} \prod_{j=0}^{N} \frac{\left(1 - (\hat{\theta}_{\{y_j, w_j\}} Z_{w_j})^2\right)}{\left(1 + \hat{\theta}_{\{y_j, w_j\}} \hat{\theta}_{\{y_j, y_{j-1}\}} Z_{w_j} Z_{y_{j-1}}\right)^2} \right| (\mathrm{diam}(T) \max(|E_1|, |E_2|, |E_3|))
$$

$$
\leq \frac{4 \, \mathrm{diam}(T)}{(2 c_4 \delta)^{4 \, \mathrm{diam}(T) + 1}}
$$

where the exponent in the denominator is $2 \, \mathrm{diam}(T) + (2 \, \mathrm{diam}(T) + 1)$ which is the worst-case bound for each of the denominators in the Hessian and maximum bound from $E_3$, respectively.

The bound for whenever $e_1 \in T_{w_k}$ is similar, except the error for the corresponding "$E_2$ term" is the same as the bound for $E_3$ above. We omit the details.

$\square$

We finally prove Lemma 4.3.

*Proof of Lemma 4.3.* We start with the simple observations that from (14) and the fact that $\hat{\boldsymbol{\theta}} \in [0, 1]^E$, and $1 + \hat{\boldsymbol{\theta}} Z_x Z_y \geq 2 c_4 \delta$ that

$$
\left| \frac{\partial^2}{\partial \hat{\theta}_e \partial \hat{\theta}_f} \ell(\hat{\boldsymbol{\theta}}; \sigma) \right| \leq \left( \frac{\hat{\theta}_e Z_x Z_v}{(1 + \hat{\theta}_e Z_x Z_y)^2} \prod_{j=1}^{N} \hat{\theta}_{\{y_j, y_{j-1}\}} \right) \prod_{j=0}^{N} \frac{\left(1 - (\hat{\theta}_{\{y_j, w_j\}} Z_{w_j})^2\right)}{\left(1 + \hat{\theta}_{\{y_j, w_j\}} \hat{\theta}_{\{y_j, y_{j-1}\}} Z_{w_j} Z_{y_{j-1}}\right)^2}
$$

$$
\leq \frac{1}{(2 c_4 \delta)^2} \prod_{j=0}^{N} \frac{\left(1 - (\hat{\theta}_{\{y_j, w_j\}} Z_{w_j})^2\right)}{\left(1 + \hat{\theta}_{\{y_j, w_j\}} \hat{\theta}_{\{y_j, y_{j-1}\}} Z_{w_j} Z_{y_{j-1}}\right)^2}.
$$

By Proposition 6.8 in (Clancy, Jr. et al., 2025a), there is some constant $C > 0$ (depending only the constants in **A1**) such that

$$
\frac{\left(1 - (\hat{\theta}_{\{y_j, w_j\}} Z_{w_j})^2\right)}{\left(1 + \hat{\theta}_{\{y_j, w_j\}} \hat{\theta}_{\{y_j, y_{j-1}\}} Z_{w_j} Z_{y_{j-1}}\right)^2} \frac{\left(1 - (\hat{\theta}_{\{y_{j+1}, w_{j+1}\}} Z_{w_{j+1}})^2\right)}{\left(1 + \hat{\theta}_{\{y_{j+1}, w_{j+1}\}} \hat{\theta}_{\{y_{j+1}, y_j\}} Z_{w_{j+1}} Z_{y_j}\right)^2} \leq \frac{C}{\delta} \qquad \text{for all } j = 0, 1, 2, \cdots, N - 1.
$$

Note that for each $j$ we have $\left(1 + \hat{\theta}_{\{y_j, w_j\}} \hat{\theta}_{\{y_j, y_{j-1}\}} Z_{w_j} Z_{y_{j-1}}\right) \geq 2 c_4 \delta$ since $\hat{\theta}_e \leq 1 - 2 c_4 \delta$ and $Z_v \in [-1, 1]$ for all $v$. It follows that

