# OpenReview forum: "Sample Complexity of Branch-length Estimation by Maximum Likelihood"
_ICML.cc/2025/Conference — ICML 2025 poster_

### Official Review · Reviewer_JvJP · 2025-03-10

**Overall Recommendation:** 3

**Summary:**

This paper focuses on the branch lengths maximum likelihood estimation problem. Arises in phylogenetic inference, this problem aims at estimating the transition probability over each edge of a bifurcating tree give repeated and independent observation of leaf node states.
The authors prove that, with the assumption of interval-bounded transition probabilities and positive correlations of adjacent states, the empirical likelihood function is convex with high probability, with polynomial many observations.
Based on this, the coordinate ascent algorithm proves to exponentially converges to the ground truth estimate.
The problem setting is very interesting and the authors give a detailed theoretical analysis, which may have a general impact on bioinformatics.

**Claims And Evidence:**

The claims in this submission are supported by clear evidence.

**Essential References Not Discussed:**

No

**Experimental Designs Or Analyses:**

There is no experimental design in this paper. I encourage the authors to verifies their conclusion empirically, which would not demand so much time on toy examples (a tree with less than 10 leaves?).

**Methods And Evaluation Criteria:**

Overall, I think this paper presents a sound methodology and gives a meaningful conclusion on the convergence rate of the coordinate ascent for branch length MLE. Below are two potential unsatisfactory points:
- The equation (2) does not not include a tree shape component, which may also affect the estimation. Have the author considered how to  perform MLE with additional freedom on tree shapes?
- The "ferromagnetic regime" in line 111 assumes positive correlations of adjacency state, this is not the case for real phylogenetics problems. How does this assumption affect your conclusion?

**Other Comments Or Suggestions:**

The authors should distinguish the use of \citet and \citep.

**Other Strengths And Weaknesses:**

This paper has a clear presentation and is easy to follow.

**Questions For Authors:**

- In many real problems, the class $\sigma_\rho \in \\{-1,1\\}$ seems not practical. For DNAs, this can be four types of nucleotides. Can your analysis easily transfer to such a case?

**Relation To Broader Scientific Literature:**

This problem is closely related to phylogenetics, or more generally, inference on black-box graphs with finite observations. The conclusion would be meaningful for broader audience.

**Theoretical Claims:**

I do not check the correctness of the proofs.

---

> ### Author Rebuttal · Authors · 2025-03-31
>
> Thank you for the review and thoughtful comments.
>
> * The equation (2) does not include a tree shape component, which may also affect the estimation. Have the author considered how to perform MLE with additional freedom on tree shapes?
>
> **Response**
> >In general, optimizing over both the edge lengths $\mathbf{\theta}$ and the tree $T$ is NP-hard [[Chor, Tuller, 2006](https://link.springer.com/chapter/10.1007/11415770_23)] and [[Roch, 2006](https://ieeexplore.ieee.org/document/1588849)]. Polynomial-time algorithms for finding the true tree under the CFN model with sufficient amount of data have been obtained [[Daskalais et al., 2011](https://arxiv.org/abs/math/0509575)] but involve ad hoc methods not used in practice; our work focuses on a common method to estimate the branch length parameters. Moreover, the MLE yields the correct pair ($\theta,T$) under the same amount of data from the CFN model *provided* that $\theta$ takes finitely many discrete values (i.e. the parameter lies on a lattice); we will include more of a discussion of this in the revised article, but see Section 2.3 in [[Roch and Sly, 2017](https://arxiv.org/abs/1508.01964)]. But there are significant roadblocks in dropping the lattice assumption in that paper (an assumption which is also made in [[Daskalais et al., 2011](https://arxiv.org/abs/math/0509575)]), and that result does not shed light on the convergence of common optimization schemes (unlike our results).
>
> * The "ferromagnetic regime" in line 111 assumes positive correlations of adjacency state, this is not the case for real phylogenetics problems. How does this assumption affect your conclusion?
>
> **Response**
> > The CFN model requires that the edge probability $p_e\in [0,1/2)$ which implies that $\theta_e = 1-2p_e \in (0,1]$. This is because the signal/character $X$ evolves as a two-state continuous-time Markov chain with generator $\displaystyle \begin{bmatrix} -1&1\\\\ 1&-1\end{bmatrix}$. In this case, it can be shown that $P(X_t = 1|X_0 = 1) = 1-\frac{1}{2}e^{-t}$ which decreases from $1$ at $t = 0$ to $1/2$ as $t\to\infty$. Hence, the ferromagnetic assumption is in fact standard in this case.
>
>
> * In many real problems, the class $\sigma_\rho\in \\{-1,1\\}$ seems not practical. For DNAs, this can be four types of nucleotides. Can your analysis easily transfer to such a case?
>
>  **Response**
>
>  >The CFN model can be used to model the nitrogenous base (purine/pyrimidine) of a nucleotide. This groups AG (purine) and CT (pyrimidine) together.
>  >There appear to be significant roadblocks for handling the $4$-state (or more general $q$-state) models. For example, for the 2-state model the gradient $\nabla \ell(\theta;\sigma|_L)$ can be represented succinctly in terms of the magnetizations (equation (13) in the article). While there are formulas for the log-likelihood $\theta_e\mapsto \ell(\theta;\sigma|_L)$ for more general models, the precise recursive formula from [[Borgs et al. 2006](https://arxiv.org/abs/math/0604366)] (or equation (29) in the article) that is used to compute the empirical gradient appears to completely breakdown. This would make understanding the population Hessian that much harder.

---

### Official Review · Reviewer_TiSt · 2025-03-13

**Overall Recommendation:** 4

**Summary:**

This work concerns the maximum-likelihood estimation in a particular model for branch-length estimation relevant in phylogenetics. This work seems to provide theoretical support for the finding that a rather naive coordinate ascent algorithm works well for this problem even though the likelihood is known to be non-concave.

**Claims And Evidence:**

This work is entirely theoretical and all the evidence consists of mathematical proofs.

**Essential References Not Discussed:**

I have nothing to add here.

**Experimental Designs Or Analyses:**

There are no experiments.

**Methods And Evaluation Criteria:**

This work is entirely theoretical. This is appropriate. But it would have been useful/insightful to add at least one numerical experiment. This would allow the authors to, e.g., compare empirical convergence rates with the theoretical convergence rates from Thm 3.3 and 3.4. Such an experiment could also give some insight into the "warm-start" requirement in Thm 3.4 (i.e., the need for starting the optimisation close to the true value).

**Other Comments Or Suggestions:**

- Line 69: converges -> converge
- Page 3, 2nd column: There is some confusion/ambiguity here as to whether $\sigma^{(j)}$ is a sample from all nodes of the tree or just the leaf nodes. That is, is $\sigma^{(j)} = {\sigma|}_L^{(j)}$? And if so, why is the 2nd notation needed?
- Lines 135, 136, 249: use natbib's citet not citep
- Line 205: maybe remind the reader of the definitions of these symbols.
- Line 220--221: (non-)convex -> (non-)concave
- Thm 3.4: is it clear what kind of norm is being used here?

**Other Strengths And Weaknesses:**

I think this paper is quite well written, motivated and structured. The contributions seem original and significant enough to warrant publication.

**Questions For Authors:**

1. Given the submission to a machine-learning conference, how does this work relate to machine learning?
2. How crucial is the "warm-start" requirement in Thm 3.4, i.e. the requirement that the optimisation is started within $O(\delta)$ distance of the true parameter. Is this a concern/problem in practice?

**Relation To Broader Scientific Literature:**

I am no expert in this area. But this work provides careful theoretical support for a finding (which seems to be known in the literature) that a naive coordinate ascent algorithm works well in the studied model for branch-length estimation despite the fact that  the likelihood is not concave in this problem. More generally, this work may provide a basis for establishing convergence rates for maximum-likelihood estimation in other models with irregular likelihood surfaces.

**Theoretical Claims:**

I did not check the proofs in detail. But I did not spot any issue which would lead me to suspect that the analysis was not carried out with a sufficient amount of mathematical rigour.

---

> ### Author Rebuttal · Authors · 2025-03-31
>
> Thank you for your comments and for pointing out the typos. We will incorporate them in the revision.
>
> 1. Given the submission to a machine-learning conference, how does this work relate to machine learning?
>
> **Response**
> > We appreciate the reviewer’s question. Maximum Likelihood Estimation (MLE) is a fundamental principle underlying many problems in machine learning. In classical statistics, the MLE landscape is often assumed to be strictly concave, ensuring robust computation and high-probability estimation of the population parameter. However, in modern machine learning applications, MLE problems are often highly non-concave, rendering conventional tools for statistical robustness and computational guarantees inapplicable.
>
> > While our work focuses on a specific MLE problem from phylogenetics, we believe our broader framework—analyzing non-concave MLE problems using the widely adopted coordinate descent algorithm—can be applied to other non-concave MLE settings in machine learning. To this end, we have formalized the following three-step approach in the Contributions section:
>
> > Step 1: Show that the population likelihood ￼ is strongly concave and smooth over some parameter space ￼ containing the true parameter ￼.
>
> > Step 2: Establish that the entries of the population Hessian vary in a Lipschitz manner with respect to the parameter.
>
> > Step 3: Demonstrate that the per-sample empirical Hessian has a uniformly bounded spectral norm almost surely.
>
> > Following this framework, researchers can establish the benign non-concavity of MLE problems. We hope our work provides a foundation for studying challenging MLE problems beyond the reach of classical methods and serves as a guideline for future research in this direction.
>
> 2. How crucial is the "warm-start" requirement in Thm 3.4, i.e. the requirement that the optimisation is started within distance of the true parameter. Is this a concern/problem in practice?
>
> **Response**
> > The warm-start requirement may not be needed and whether or not it is needed is indeed a very interesting topic for future research; however, in general, greedy coordinate maximization should not always find the global maximizer of a non-concave objective function. Indeed, there is a classical counterexample by [[Powell, 1973](https://link.springer.com/article/10.1007/BF01584660)] on coordinate maximization failing to even converge to a stationary point of a smooth objective. The likelihood function in our setting is known to have exponentially many critical points in terms of the size of the tree, so it is highly likely that the success of coordinate maximization cannot be warranted with arbitrary initialization. Our analysis circumvents this issue by placing the initialization sufficiently inside the "good box" $\hat{\Theta}_{0}(\delta)$ (please refer to Fig. 1 in the paper) so that, with high probability, the coordinate maximization algorithm can only experience strongly convex landscape without knowing how large it is.
>
> 3. Numerical example.
>
> **Response**
> > The population landscape paper by Clancy et al. that we cite does not contain precise control over how small the parameter $\delta$ needs to be and so we cannot guarantee that any numerical example would relate to the results we prove; however, we will certainly try to conduct numerical experiments. If any are successful, we will include them in the updated version.

---

### Official Review · Reviewer_9aGs · 2025-03-18

**Overall Recommendation:** 5

**Summary:**

The paper provides analysis of optimization landscape of the MLE problem in phylogenetics under the Kesten-Stigum (KS) regime. As a corollary, they obtain quantitative results for consistency of the MLE and convergence rate for coordinate descents, which are often used in practice.

**Claims And Evidence:**

The paper is theoretical in nature. All claims written in theorem formats are carefully written and are correct, as far as the reviewer has investigated the proof. However, I believe that some out-of-theorem-format claims could be written a bit more carefully.

For instance, the whole paper assumes that we are working in the KS regime, which is a major assumption, since the KS regime is known to be 'easier' for phylogenetics MLE.

**Essential References Not Discussed:**

Roch and Sly's "Phase transition in the sample complexity of likelihood-based phylogeny inference" shows that the sample complexity in the KS regime can be proven to be very small (logarithmic in the number of taxa). Given that the proposed name of the paper is about 'sample complexity', a more direct comparison with Roch and Sly's paper is in order. Note that the optimization landscape results are still novel, just that the sample complexity results need more comparisons to be placed in existing literature.

Clancy, Ryu and Roch earlier this year also has a preprint on "Likelihood landscape of binary latent model on a tree", which also analyze the optimization landscape of the same problem. Given that this paper also derives their main theorems from a landscape study, I believe that a more in depth discussion of how this results differ from that of Clancy et al. paper is warranted

**Experimental Designs Or Analyses:**

Experiments are small-scaled and mostly toyish, which is expected and appropriate for a heavily theoretical paper.

**Methods And Evaluation Criteria:**

The suggested methods to analyze optimization landscape, starting with population Hessian bounds and translate to empirical results via a concentration inequality is well known and correct. The paper is novel in the particular optimization object that is being applied, as well as a new uniform concentration inequality, which might be of independent interest.

**Other Comments Or Suggestions:**

See strengths and weaknesses.

**Other Strengths And Weaknesses:**

The paper is overall interesting and addresses a timely problem in analyzing loss landscape of optimization that seems to be well-solved by  descent-based algorithms. The theoretical analysis is sound. My biggest concern is that the related work session is not well down, with very few direct comparisons to relevant literature. Indeed I think the paper has substantial overlaps with Clancy, Ryu and Roch's preprint on "Likelihood landscape of binary latent model on a tree" (which the paper did cited) as both analyze the landscape of MLE and arrive at a regularity condition. The second improvement that the paper can make is to clarify very early on, and in the abstract, that the paper is assuming the KS regime, which makes MLE tree inference (which is NP hard in general) much easier in time complexity. The third improvement would be a more detailed discussion on this crucial regime (perhaps in the appendix) so that the proof intuition can be derived quicker. I am giving the paper a borderline acceptance score, simply to do substantial overlap with previous work, which can be seen as concurrent work, but would be willing to raise it higher if my concerns are addressed.

**Questions For Authors:**

See strengths and weaknesses.

**Relation To Broader Scientific Literature:**

MLE is one of the most used methods in tree inference in phylogenetics analysis due to its nice statistical guarantees (e.g. low sample complexity, statistical consistency, etc.). However, it is also NP hard to compute and practical applications of the method employs local heuristics, which lacks theoretical understandings. This paper is part of a recent effort to understand the success of descent-based heuristics used in practice.

**Theoretical Claims:**

I have checked the proof strategies and I believe that they are sound. There might still be typos in the proof since I did not check the proof line by line, but the overall idea is correct.

---

> ### Author Rebuttal · Authors · 2025-03-31
>
> Thank you for your thoughtful comments and remarks.
>
> * My biggest concern is that the related work session is not well down, with very few direct comparisons to relevant literature. Indeed I think the paper has substantial overlaps with Clancy, Ryu and Roch's preprint on "Likelihood landscape of binary latent model on a tree" (which the paper did cited) as both analyze the landscape of MLE and arrive at a regularity condition.
>
>
>
> **Response:**
> > We will flesh out the related works section and include a discussion on the paper of Roch and Sly you mention and what distinguishes our results from theirs. For example, they make a crucial additional assumption that the parameter lives on a lattice while we do not need that assumption. Moreover Roch and Sly show that, given sufficiently many samples, the likelihood is maximized at the true discretized parameters (including the tree parameter) with high probability, but they do not address the question of the convergence of standard optimization algorithms - the focus of our work.
>
>
> > In the paper by Clancy et al. that you mention, the authors establish eigenvalue bounds of the population Hessian in an $L^\infty$ ball around the true parameter $\theta^*$. We indeed use this result in our paper - this is Step 1 in our three-step program described on page 2. Our paper is a demonstration that a commonly used optimization algorithm for MLE with non-concave likelihood landscapes does work for this particular model. Moreover, the paper by Clancy et al. is not sufficient in itself to guarantee that the empirical log-likelihood is strongly concave and smooth uniformly in some region. Standard matrix concentration results applied to the empirical Hessian would only allow for high-probability statements of the empirical Hessian for a fixed $\theta$ (or finitely many $\theta$'s). To improve this to uniform, we need stronger matrix concentration results and (usable) a.s. bounds on the Lipschitz constant of the Hessian to obtain high-probability and uniform control over the fluctuations of the empirical Hessian about its mean (this is our Appendix E). This, combined with our uniform matrix Berstein's inequality, give us high-probability control of the empirical log-likelihood function.

---

### Decision · Program_Chairs · 2025-05-01

**Decision:**

Accept (poster)

**Comment:**

This paper analyzes the optimization landscape of the MLE problem for phylogenetic tree branch length estimation under the Kesten-Stigum regime; the main focus is on explaining the success of coordinate ascent methods.

The reviewers noted some weaknesses, including insufficient discussion of related work, the realism of assumptions, and the lack of numerical experiments. Additionally, the scope of the work, while technically solid, may be somewhat limited for a broad machine learning venue like ICML. However, the reviewers found the theoretical contributions to be sound and novel.

The authors are encouraged to take into account the reviewers’ suggestions; this will strengthen the manuscript.